

# A compact Incoherent Broadband Cavity Enhanced Absorption Spectrometer (IBBCEAS) for trace detection of nitrogen oxides, iodine oxide and glyoxal at sub-ppb levels for field application

Albane Barbero[1], Camille Blouzon[1], Joël Savarino[1], Nicolas Caillon[1], Aurélien Dommergue[1], and Roberto Grilli[1]

[1]Univ. Grenoble Alpes, CNRS, IRD, Grenoble INP*, IGE, 38000 Grenoble, France
*Institute of Engineering Univ. Grenoble Alpes

**Correspondence:** Roberto Grilli (roberto.grilli@cnrs.fr)

**Abstract.** We present a compact, affordable and robust instrument based on Incoherent Broadband Cavity Enhanced Absorption Spectroscopy (IBBCEAS) for simultaneous detection of $NO_x$, IO, CHOCHO and $O_3$ in the 400 – 475 nm wavelength region. The instrument relies on the injection of a high-power LED source in a high-finesse cavity (F $\sim$ 36,100), with the transmission signal be detected by a compact spectrometer based on a high-order diffraction grating and a CCD camera. A minimum

detectable absorption of $1.8 \times 10^{-10}$ cm$^{-1}$ was achieved within $\sim$ 22 minutes of total acquisition, corresponding to a figure of merit of $7.5 \times 10^{-11}$ cm$^{-1}$ Hz$^{-1/2}$ per spectral element. Due to the multiplexing broadband feature of the setup, multi-species detection can be performed with simultaneous detection of $NO_2$, IO, CHOCHO, and $O_3$ achieving ultimate detection limits of 9, 0.3, 8 ppt and 40 ppb ($1\sigma$) within 22 min of measurement, respectively (half of the time spent on the acquisition of the reference spectrum in absence of absorber, and the other half on the absorption spectrum). The implementation on the

inlet gas line of a compact ozone generator based on electrolysis of water allows the measurement of $NO_x$ (NO + $NO_2$) and therefore an indirect detection of NO with detection limits for $NO_x$ and NO of 12 and 21 ppt ($1\sigma$), respectively. The device has been designed to fit in a 19", 3U rack-mount case, weights 15 kg and has a total electrical power consumption < 300 W. The instrument can be employed to address different scientific objectives such as better constraint the oxidative capacity of the atmosphere, study the chemistry of highly reactive species in atmospheric chambers as well as in the field, and looking at the

sources of glyoxal in the marine boundary layer to study possible implications on the formation of secondary aerosol particles.

## 1 Introduction

Free radicals are controlling the oxidative capacity of the atmosphere and therefore contribute to the upholding of its chemical balance. With their unpaired valence electron, they are highly chemically reactive, and are therefore considered the "detergents" of the atmosphere (Monks, 2005; Monks et al., 2009). Even if present at extremely low concentrations, radicals are constantly

formed by photochemical and combustion processes. They may be removed from the atmosphere by biological uptakes, dry and wet deposition, and chemical reactions (Finlayson-Pitts and Pitts, 2000). Free radicals in the troposphere such as nitrogen oxides ($NO_x$), hydroxyl radical (OH), peroxy radicals ($HO_2$, $RO_2$) and halogen oxides (BrO and IO), can be found at mixing



ratios (i.e. mole fractions) ranging from less than one part per trillion ($10^{-12}$ mol mol$^{-1}$ or ppt) up to a few parts per million ($10^{-6}$ mol mol$^{-1}$ or ppm) in the atmosphere (Wine and Nicovich, 2012). Measuring their concentration and dynamic variability

in different atmospheric environments is key for addressing specific questions regarding air quality, the oxidative state of the atmosphere, the ozone budget, aerosol nucleation, as well as carbon, nitrogen and sulfur cycles. The understanding of the complex interactions involving those species has led to numerous investigations during the past decades. Especially, nitrogen oxides ($NO_x$ = NO and $NO_2$), have a direct impact on air quality and climate change. In presence of volatile organic compounds (VOCs) and under solar radiation, nitrogen oxides stimulate ozone ($O_3$) formation in the troposphere. $NO_x$ also plays an

important role in rain acidification and ecosystems eutrophication by its transformation in nitric acid ($HNO_3$) (Jaworski et al., 1997; Vitousek et al., 1997). Finally, $NO_x$ contribute to the formation of particulate matter in ambient air and to the aerosol formation leading to clouds formation (Atkinson, 2000). The $NO_2$ mixing ratio in the troposphere ranges from a few tens of ppt in remote areas to hundreds of ppb ($10^{-9}$ mol mol$^{-1}$) in urban atmospheres (Finlayson-Pitts and Pitts, 2000). Being able to measure such species in situ, at low levels and at a time scale compatible with its reactivity (i.e. in min) is challenging

and puts stringent constraints on the instrument sensitivity, time response, energy consumption and compactness. Among the various techniques that have so far been developed, ChemiLuminescence Detection (CLD) (Maeda et al., 1980; Ryerson et al., 2000), Long-Path Differential Optical Absorption Spectroscopy (LP-DOAS) (Lee et al., 2005; Pikelnaya et al., 2007; Lee et al., 2008), and Multi-AXis Differential Optical Absorption Spectroscopy (MAX-DOAS) (Platt and Perner, 1980; Sinreich et al., 2004; Wagner et al., 2010) have been used to detect nitrogen species and halogen oxides. The CLD technique, using the

chemiluminescence reaction occurring between $O_3$ and NO after the reduction of $NO_2$ into NO, is widely used for air quality measurements with sensitivities better than 100 ppt (Ryerson et al., 2000). Nevertheless, the interferences in the reduction of $NO_2$ to NO with other species (i.e. HONO, $HNO_3$) and the sensitivity to environmental conditions (temperature and humidity) leave uncertainties on absolute mixing ratio measurements (Grosjean and Harrison, 1985; Williams et al., 1998). The MAX-DOAS technique has been used to measure BrO and $NO_2$ by making use of the characteristic absorption features of gas

molecules along a path in the open atmosphere (Leser et al., 2003). Although MAX-DOAS is relatively simple to deploy, the data analysis makes it a complex approach for in situ field measurements due to the influence of clouds on the radiative transfer which alters the path length of light (Wittrock et al., 2004; Rozanov and Rozanov, 2010). While in LP-DOAS the optical path length is known, the signal degradation due to the environment (clouds, rain, wind) remains of importance for the data retrieval and results are integrated over the long path leading to a limited spatial resolution (Chan et al., 2012; Pohler et al., 2010).

Compact, high sensitive and point-source measurements may be achieved using cavity enhanced techniques such as Cavity Ring Down Spectroscopy (CRDS) and Cavity Enhanced Absorption Spectroscopy (CEAS) (Atkinson, 2003). The potential of the CRDS for accurate, sensitive and rapid measurements in a compact and transportable instrument has already been demonstrated (Fuchs et al., 2009; Brown et al., 2002), e.g. Fuchs et al. (2009) reached a sensitivity of 22 ppt for $NO_2$ within 1 s of integration time using the CRDS technique (Fuchs et al., 2009). Incoherent Broadband Cavity Enhanced Absorption

Spectroscopy (IBBCEAS) (Fiedler et al., 2003) is a simple and robust technique for in situ field observations. Different sources and wavelength regions have been used for the detection of $NO_2$ leading do different performances: Venables et al. (2006) were able to detect simultaneously $NO_3$, $NO_2$, $O_3$ and $H_2O$ in an atmospheric simulation chamber with a sensitivity of tens





of ppb for $NO_2$ (Venables et al., 2006) ; Gherman et al. (2008) reached $\sim 0.13$ ppb and $\sim 0.38$ ppb for HONO and $NO_2$

in a 4 m$^3$ atmospheric simulation chamber between 360 and 380 nm (Gherman et al., 2008) ; Triki et al. (2008) used a red

LED source centered at 643 nm reaching a sensitivity of 5 ppb (Triki et al., 2008) ; Langridge et al. (2006) developed an

instrument with a blue light emitting diode (LED) centered at 445 nm allowing detection limits ranging from 0.1 to 0.4 ppb

(Langridge et al., 2006) ; Ventrillard-Courtillot and colleagues reached 600 ppt detection limit for $NO_2$ with a LED centered at

625 nm (Ventrillard-Courtillot et al., 2010) ; while Thalman and Volkamer reported a detection limit of 30 ppt within 1 min of

integration time (Thalman and Volkamer, 2010). More recently, Min et al 2016 proved a sensitivity of 80 ppt in 5 s of integration

at 455 nm using a spectrometer with a thermelectric cooled CCD camera and very higher reflective mirrors (Min et al., 2016).

This non-exhaustive list of works underline the need of robust, compact and transportable instruments also allowing direct

multi-species detection and low detection limits for applications in remote areas such as Antarctica, where the expected mixing

ratio of $NO_2$ could be as low as a few tens of ppt. Fuchs and colleagues, during the $NO_3$Comp campaign at the SAPHIR

atmospheric simulation chamber, demonstrated the potential of theses optical techniques to compete with the CLD instruments

as routine measurements of $NO_2$ concentrations in the future (Fuchs et al., 2010). The present paper describes a compact

and affordable instrument based on the IBBCEAS technique, allowing the simultaneous detection of nitrogen dioxide, iodine

oxide, glyoxal and ozone ($NO_2$, IO, CHOCHO and $O_3$), with detection limits of 9, 0.3, 8 ppt and 40 ppb ($1\sigma$), respectively,

for a measurement time of 22 min (half of the time spent on the acquisition of the reference spectrum in absence of absorber,

and the other half on the absorption spectrum). The four species are directly detected by a broadband blue light emitting

diode centered at 445 nm. The wavelength region was selected in order to optimize the detection of $NO_2$. Direct detection of

NO is only possible in UV region for wavelengths around 226 nm (Dooly et al., 2008) or in the mid-infrared region at 5.3

$\mu$m (Richard et al., 2018). Wavelengths difficult to achieve with LED technology. Here, an indirect measurement is proposed

which relies on the oxidation of NO to $NO_2$ under a controlled excess of $O_3$. The sum of NO and $NO_2$ is therefore measured

leading to a supplemental indirect measurement of NO if concentration of $NO_2$ is also monitored. The field deployment for

the measurements of $NO_2$ and $NO_x$ consists of two twin instruments, IBBCEAS-$NO_2$ and IBBCEAS-$NO_x$, the later equipped

with an ozone generator system.


## 2 Method

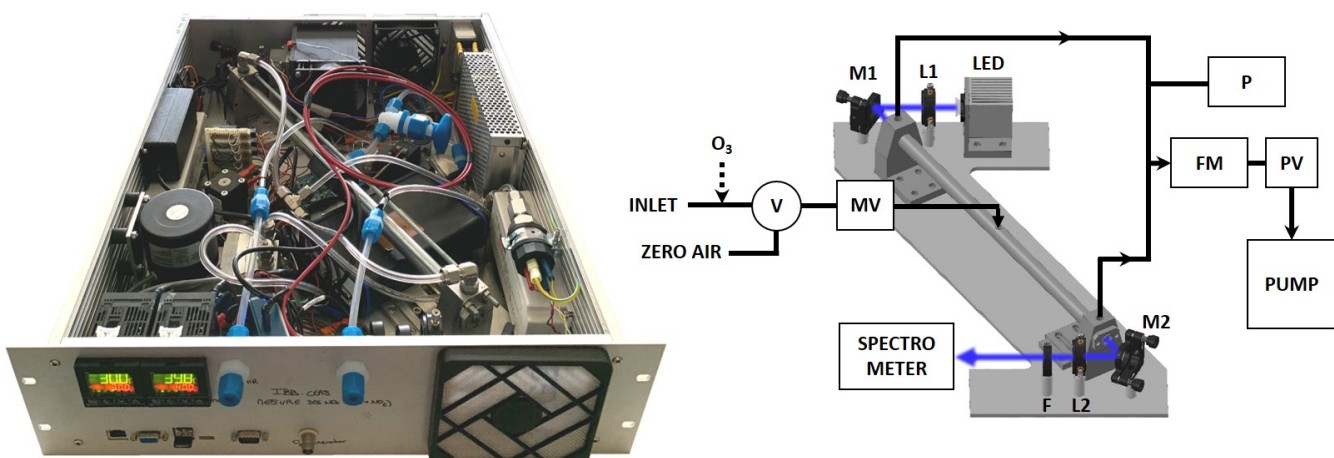

**Figure 1.** (left) A picture of the instrument mounted on a 19", 3U rack-mount case. (right) Schematic of the instrument. The light from the LED is collimated by lens L1 and injected into the cavity. The exiting light is then collimated with lens L2, and injected into the spectrometer. M1 and M2 are steering mirrors and F is an optical filter. The gas line is composed of a pump, a pressure sensor P, a flow meter FM, and a proportional valve PV. At the inlet, a 3-way 2-position valve in PTFE, V, is used to switch between the sample and zero-air. A manual PFA needle valve MV, is used to fix the flow rate. An ozonizer can be inserted in the inlet line for NO$_x$ measurements.

In IBBCEAS a broadband incoherent light source is coupled to a high-finesse optical cavity for trace gas detection. A picture of the spectrometer and a schematic diagram of the setup is shown in Figure 1. In the present study, the broadband light source consisted of a high-power LED Luminus SBT70 allowing $\sim$ 1 W of optical power to be injected into the resonator. A thermoelectric (TEC) Peltier cooler (ET-161-12-08-E) and a fan/heatsink assembly were used to directly evacuate outside of the instrument up to $\sim$ 75 W of thermal heat from the LED. A temperature regulator (RKC RF100) with a PT100 thermistor was used to stabilize the LED temperature at $\pm$ 0.1 °C. The LED spectrum was centered at 445 nm with 19 nm FWHM (Full Width at Half Maximum) which covers the main absorption features of NO$_2$, IO, CHOCHO and O$_3$. For better collimation of the LED spatially divergent emission (7 mm$^2$ surface), a dedicated optic (Ledil HEIDI RS) was used and coupled with a 25 mm focal lens (L1, Thorlabs, LA1951-A). The high-finesse optical cavity was formed by two half-inch diameter high reflectivity mirrors (maximum reflectivity at 450 nm $\geqslant$ 99.990 $\pm$ 0.005 %, Layertec, 109281) separated by a 41.7 cm-long PFA tube (14 mm internal diameter, 1 mm thick) hold by an external stainless-steel tube. Both mirrors were pre-aligned and glued with Torr Seal epoxy glue on removable stainless-steel supports which were then screwed on the cavity holders. This enables the easy cleaning





of the mirrors when required and also the removal of the cavity tube to perform open-cavity measurements, which is of interest
      for the detection of the highly reactive IO radical. Behind the cavity, a Thorlabs FB450-40 filter was used in order to remove the
      broadband component of the radiation sitting outside the highly reflective curve of the cavity mirrors. The radiation is focused
      on an optical fiber (FCRL-7UV100-2-SMA-FC) using a 40 mm focal lens (L2, Thorlabs, LA1422-A). The optical fiber input
      was composed of 7 cores in a round shape pattern on the collecting side, whereas, at the fiber end, on the spectrometer side, the
cores were assembled in a line for better matching the 100 $\mu$m slit at the spectrometer. The spectrometer (Avantes, AvaSpec
      ULS2048L) was composed of a diffraction grating (2,400 lines mm$^{-1}$) and 2,048 pixels charge-coupled device (CCD). The
      resolution of the spectrometer was $0.54 \pm 0.10$ nm. All the optics including the cavity were mounted on a Z-shaped 8-mm
      thick aluminum board fixed on the rack using cylindrical dampers (Paulstra). On the board, four 5 W heating bands and one
      PT100 sensor were glued, and a second RKC module used to regulate its temperature. The board therefore acts as a large
radiator inside the instrument, allowing to minimize internal thermal gradients and thermalize the instrument. Air circulation
      from outside is ensured by an aperture at the front and a fan placed at the back wall of the instrument (Figure 1). The gas line
      system was composed of a manual PFA needle valve (MV) and a 3-way 2-position PTFE valve, V (NResearch, 360T032) at
      the entrance ; while a proportional valve PV (Burkert, 239083), a flowmeter F (Honeywell, HAFUHT0010L4AXT), a pressure
      sensor P (SLS ATM.ECO) and a diaphragm pump (KNF, N 816 AV.12DC-B) were placed after the cavity. The entire line was
made of ¼" PFA tubing which was found to be least lossy for the transport of highly reactive species (Grilli et al., 2012). The
      pump provided a constant flow that can reach 11 L min$^{-1}$ at the end of the gas line while a constant pressure in the cavity
      was obtained by a PID regulator on the proportional valve. A data acquisition card (National Instruments, USB 6000) was
      interfaced to read the analogue signal from the pressure sensor, while a microcontroller (Arduino Due) drived the proportional
      valve. The manual valve at the entrance allowed to tune the flow rate. At the inlet, a 3-way 2-position PTFE valve allowed to
switch between the gas sample and zero-air mixture for acquiring a reference spectra in the absence of absorption. Zero-air
      was produced by flowing outdoor air through a filtering system (TEKRAN, 90-25360-00 Analyzer Zero Air Filter). 9 $\mu$m
      particle filters were also placed in the inlet lines (reference and sample) for preventing optical signal degradation due to Mie
      scattering as well as a degradation of the mirror reflectivity for long term deployment. The air flow was introduced at the center
      of the cavity and extracted at both ends of the cavity. The optimal cavity design was selected by running SolidWorks flow
simulations at flow rates between 0.5 and 1 L min$^{-1}$ (for more details see supplementary informations - SI). Cavity mirrors
      were positioned in order to maximize the sample length $d$ (therefore minimizing dead-space) while avoiding that air flow
      would hit the mirror surfaces leading to a gradual degradation of the cavity finesse over time. All the components fit in a 19",
      3U aluminum rack-mount case, have a total weight of 15 kg and a total electrical power consumption < 300 W. Instrument
      interface, measurements and data analysis are performed automatically, without the intervention of an operator, by dedicated
LabView software. Instrument calibrations, however, must be performed by an operator on a regular basis.





## 3 Spectral fit

The absorption spectrum is calculated as the ratio between the spectrum of the light transmitted through the cavity without a sample, $I_0(\lambda)$, and with a sample in the cavity, $I(\lambda)$. It is expressed as the absorption coefficient (in units of cm$^{-1}$) by the following equation (Fiedler et al., 2003):

$$\alpha(\lambda) = (\frac{I_0(\lambda)}{I(\lambda)} - 1)(\frac{1 - R(\lambda)}{d}) \tag{1}$$

where $R(\lambda)$ is the wavelength dependent mirror reflectivity and $d$ the length of the sample inside the cavity. Equation (1) is derived from the Beer-Lambert's law and applied to light in an optical resonator (Ruth et al., 2014). The light transmitted through the optical cavity is attenuated by different processes such as absorption, reflection and scattering of the mirror substrates and coating, as well as losses due to the medium inside the cavity. The losses of the cavity mirrors are assumed to be

constant between the acquisition of the reference and the sample spectrum. Mie scattering is minimized with a particle filter in the gas inlet, while Rayleigh scattering losses were calculated to be $2.55 \times 10^{-7}$ cm$^{-1}$ at 445 nm at 25 °C and 1 atm (Kovalev and Eichinger, 2004) and thus negligible with respect to the cavity losses normalized by the cavity lengh ($\frac{1-R}{d} = 2.09 \times 10^{-6}$ cm$^{-1}$). Therefore, the light transmitted through the cavity is mainly affected by the absorption of the gas species, which leads to well-defined absorption spectral features, $\alpha_i(\lambda)$, that are analyzed in real time by a linear multicomponent fit routine.

Experimental absorption spectra of the species $i$ ($i$ = NO$_2$, IO, CHOCHO and O$_3$) have been compared with literature cross section data accounted for the gas concentration, experimental conditions of temperature and pressure, and convoluted with the spectrometer instrumental function. Those experimental spectra are then used as reference spectra for the fit.

$$\alpha(\lambda) = \sum_i \sigma_i(\lambda)c_i + p(\lambda) \tag{2}$$

A fourth order polynomial function, $p(\lambda) = a_0 + a_1\lambda + a_2\lambda^2 + a_3\lambda^3 + a_4\lambda^4$, is added to the absorption coefficient equation (2)

to adjust the spectral baseline and account for small changes between the reference and the sample spectra. The transmitted light intensity, as well as the optical absorption path, will be modulated by the shape of the mirror reflectivity curve. Therefore, the later should be defined in order to retrieve the correct absorption spectrum recorded at the cavity output.

## 4 Calibration, performance and multi-species detection

### 4.1 Calibration

Washenfelder and coworkers (Washenfelder et al., 2008) described a procedure for retrieving the mirror reflectivity curve by taking advantage of a different Rayleigh scattering contribution to the cavity losses while the measuring cell was filled with different bulk gases (eg. Helium versus air or nitrogen). In this work we propose an easier approach consisting of using a trace gas at a known concentration (in this case NO$_2$, since O$_3$ spectrum is less structured, IO is highly reactive and CHOCHO is not easy to produce at a known concentration) and its literature cross-sections (Vandaele et al., 1998) for retrieving the wavelength

dependent reflectivity curve. The shape of the reflectivity curve is first approximated with a fourth order polynomial function,





$p(\lambda)$, from the theoretical curve provided by the manufacturer. Then, reflectivity values over the broadband spectral region are deduced experimentally by adjusting a constant parameter (maximal reflectivity) and polynomial parameters (shape) of the reflectivity curve in order to best match the literature $NO_2$ spectrum. See SI for further details. Figure 2(a) shows the resulting reflectivity curve and the transmitted light through the cavity and the optical filter. The maximum reflectivity achieved with

both mirrors given by the calibration procedure is 99.9913 % which leads to an effective optical path length of $\sim$ 4.8 km and a cavity finesse (F = $\frac{\pi \sqrt{R}}{(1-R)}$) of $\sim$ 36,100. While the shape of the mirror reflectivity curve is determined once and for all, its offset is slightly adjusted after each mirror cleaning, by flushing in the cavity a known concentration of $NO_2$. The spectral emission of the LED centered at 445 nm is well suited also for the detection of IO, CHOCHO and $O_3$, which are other key species in atmospheric chemistry. For the field measurements of $NO_2$ and $NO_x$, two twin instruments named IBBCEAS-$NO_2$

and IBBCEAS-$NO_x$ are deployed, with the later equipped with an ozone generator on the gas inlet line. At this wavelength region water vapor also absorbs and is accounted in the spectral fit analysis. However, the absorption of oxygen dimer is not required in the fit routine since the absorption feature will be present in the reference ($I_0$) as well as in the a absorption ($I$) spectra. In Figure 2(b) simultaneous detection of species $NO_2$, IO and $O_3$ is reported. Ozone, at 26.5 ppm, was produced by water electrolysis as described in Section 4.4, 175.6 ppb of $NO_2$ were provided by a permeation tube, and 389.7 ppt of IO were

generated by photochemical reaction of sublimated iodine crystals and ozone in the presence of radiation inside the cavity. For this spectrum, the light transmitted was integrated for 350 ms on the CCD and averaged over 1000 spectra, yielding to a $1\sigma$ standard deviation of the residuals, (Figure 2(c)), of $4 \times 10^{-8}$ cm$^{-1}$. Figure 2(b) shows the experimental spectra (black trace), the fit result (red trace) and contributions from each species which have been included in the spectral fit.

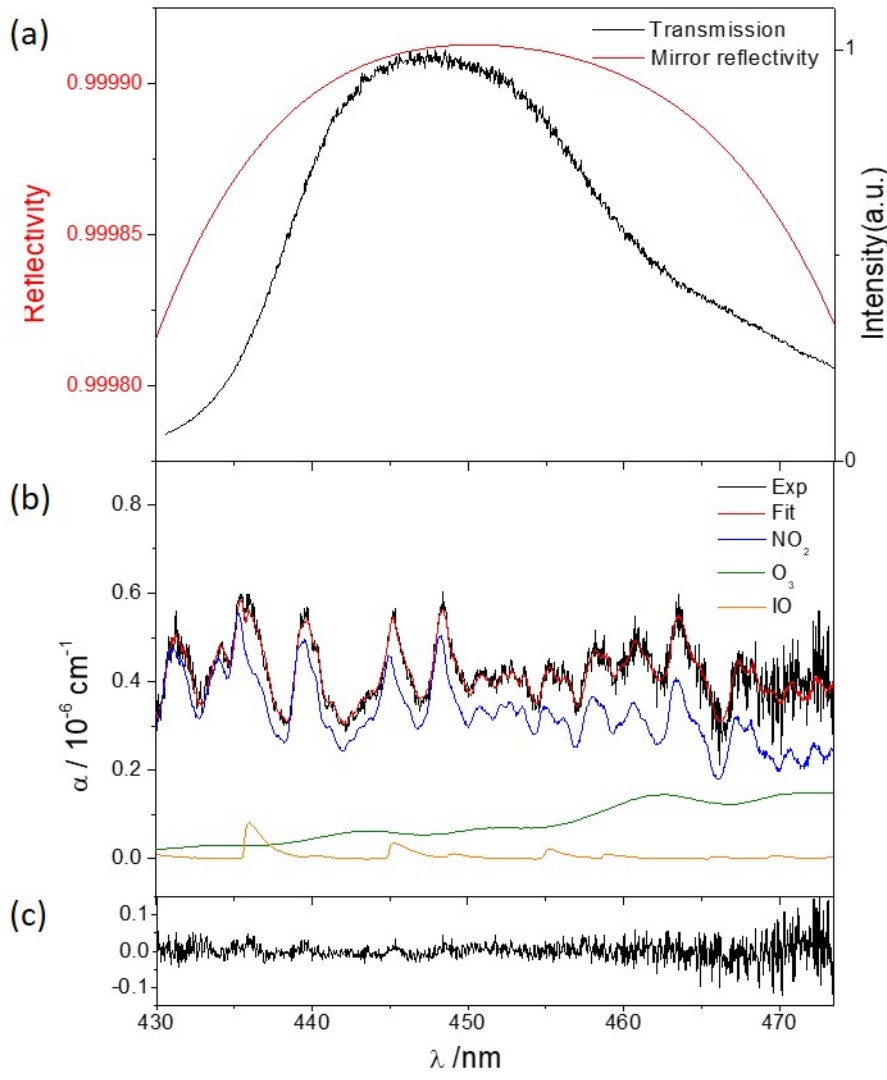

**Figure 2. (a) The mirror reflectivity curve (red) in comparison with the spectrum of the LED light transmitted by the cavity and the optical filter for a single acquisition of 350 ms. (b) In black, an example of an experimental spectrum of NO$_2$, IO and O$_3$ at concentration of 175.6 ppb, 389.7 ppt and 26.5 ppm, respectively ; in red, the multi-species spectral fit ; and in blue, orange and green the absorptions of the different species. (c) The residual of the experimental fit with a 1$\sigma$ standard deviation of $4 \times 10^{-8}$ cm$^{-1}$ after 1000 averages.**





## 4.2 Instrumental inter-comparison and calibration

As standard gas, a $NO_2$ bottle from Air Liquide ($NO_2$ in $N_2$ announced at $1.00 \pm 0.05$ ppm ($2\sigma$)) was used to calibrate the IBBCEAS instruments. To confirm the right amount of $NO_2$ in the bottle, the later was first calibrate against a CLD instrument (ThermoFisher™, 42iTL trace analyzer calibrated to NIST traceable standards by the manufacturer just before the experiments). The $NO_2$ concentration in the bottle was measured at $577.4 \pm 2.3$ ppb. The large discrepancy with respect to the value provided by the manufacturer probably comes from the losses due to the presence of the gas regulator. This gas bottle

was then used as local standard for the calibration of the IBBCEAS systems. To confirm the calibration process as well as the stability of the instrument within a greater range of concentrations, two inter-comparisons of the IBBCEAS with two different CLD instruments (ThermoFisher™, 42i $NO_x$ analyser and ThermoFisher™, 42iTL $NO_x$ trace analyser) were performed in outdoor air over 39 and 12 hours, respectively. Results are reported in Figure 3. The experiments took place at the Institute of Geosciences of the Environment (IGE) in Saint Martin d'Hères, France. The IGE is located in the university campus, $\sim 1$

km from the city center of Grenoble and $\sim 300$ m from a highway. Ambient air was pumped simultaneously from the same gas line by the instruments at flow rates of 1.0 and 0.5 L min-1 for the IBBCEAS and the CLD instrument (ThermoFisher™, 42i $NO_x$ analyser), respectively. The measurements were conducted from 6 pm on Saturday $29^{th}$ of September until 9 am on Monday $1^{st}$ of October 2018 (Figure 3(a)). On Saturday $29^{th}$ of September evening the $NO_2$ peak occurs at slightly later time than normally expected (from 8 pm to midnight). This may be due to the fact that during Saturday night, urban traffic can be

significant until late, but also due to severe weather conditions prevailing at this time, with a storm and lightnings known to be a major natural source of $NO_x$ (Atkinson, 2000). For the second experiment shown in Figure 3(b), ambient air was pumped at flow rates of 1.0 and 0.8 L min$^{-1}$ for the IBBCEAS and the CLD trace instrument (ThermoFisher™, 42iTL $NO_x$ trace analyser), respectively. The measurements were conducted from 8 pm on Thursday $18^{th}$ of July until 8 am on Friday $19^{th}$ of July 2019. Both instruments showed the expected variability from an urban environment with an increase of $NO_2$ in the

evening and morning due to photochemical processes and anthropogenic activities (i.e mainly urban traffic).

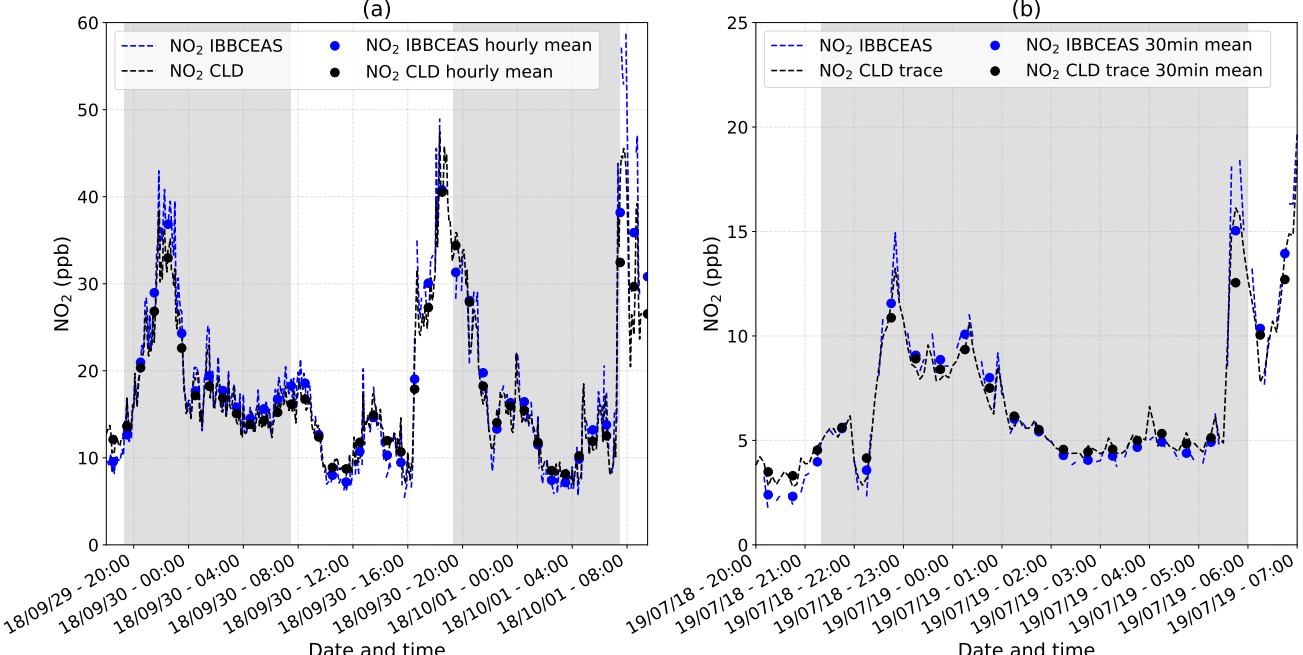

**Figure 3. (a) A 39h-long intercomparison of the IBBCEAS instrument and the commercial CLD instrument (ThermoFischer™, 42i analyzer) on the NO$_2$ detection in outdoor urban area performed in September 2018. The plot reports continuous (dashed lines) and hourly (dots) average data for both techniques. The grey area corresponds to night time period. (b) A 12h-long intercomparison of the IBBCEAS instrument and the commercial CLD trace instrument (ThermoFischer™, 42iTL trace analyzer) on the NO$_2$ detection in outdoor urban area performed in July 2019. The plot reports continuous (dashed lines) and 30 minutes (dots) average data for both techniques. The grey area corresponds to night time period.**

The correlation plot, based on data of all instruments, (Figure 4(a)), shows good linearity with a slope of $1.064 \pm 0.118$ and a correlation coefficient $R^2 = 0.960$ with measurements averaged over 5 minutes. In order to perform linearity tests, the previous NO$_2$ bottle from Air Liquide was used and diluted with a zero-air line to produce NO$_2$ at concentrations of 0, 18.2, 80.8 and 139.7 ppb. Figure 4(b) shows the good linearity of the IBBCEAS instrument with a slope of $0.968 \pm 0.019$ and a correlation factor of $R^2 = 0.996$. While the system measures NO$_2$ directly, the CLD technique applies an indirect measurement of NO$_x$ from the oxidation of NO through a catalyzer, then in CLD, the NO$_2$ mixing ratio is obtained by the subtraction of the NO signal to the total NO$_x$ signal. The discrepancies observed at low concentrations (< 5 ppb) between the two techniques maybe due to the fact that the measurement from the CLD could actually corresponds to NO$_y$, leading to an overestimation of the NO$_2$ concentration.

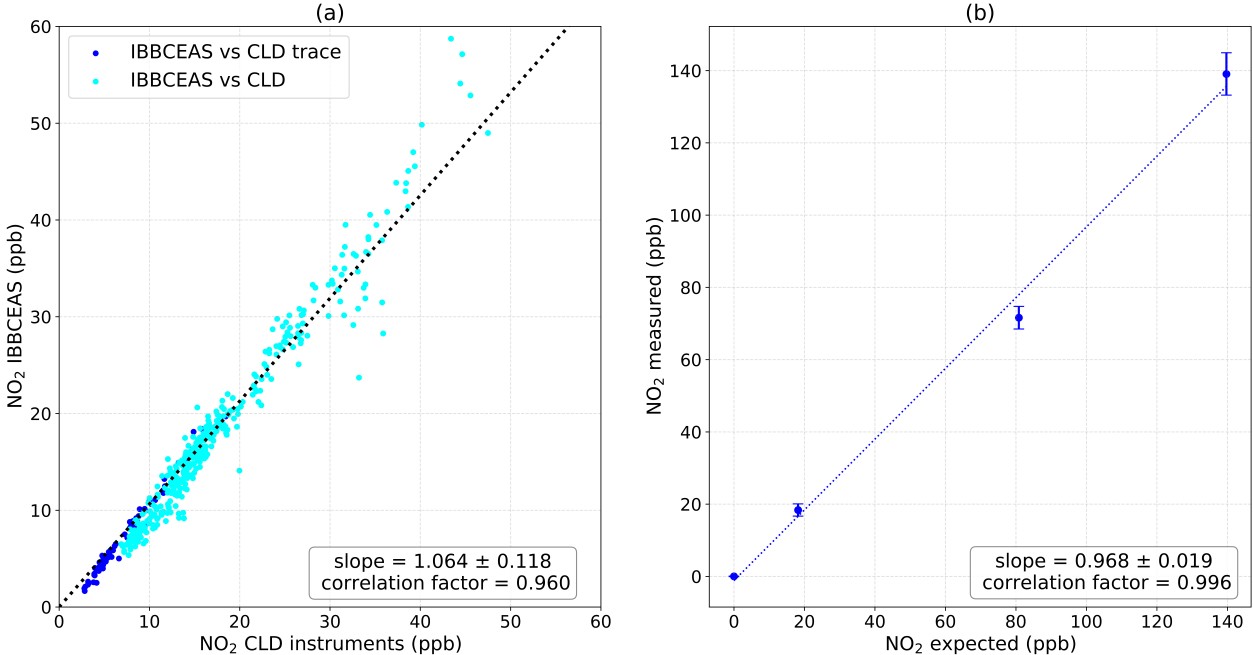

**Figure 4. (a)** A linear correlation was obtained with a slope of $1.064 \pm 0.118$ and a correlation coefficient $R^2 = 0.960$ between the IBBCEAS system and the ThermoFisher instruments. **(b)** Results of the system's calibration using a $NO_2$ bottle from Air Liquide in a dilution line. A linear correlation was obtained with a slope of $0.968 \pm 0.019$ and a correlation coefficient of $R^2 = 0.996$

## 4.3 Performance

### 4.3.1 Instrument sensitivity and long-term stability

In remote area such as East Antarctica, $NO_2$ ranges from a few tens to a few hundreds of ppt (50 - 300 ppt) (Frey et al., 2013, 2015). This requires the sensitivity of a field instruments to be at the level of a few tens of ppt. Due to the low signal-to-noise ratio of the spectrometer, a single acquired spectrum (with an integration time ranging between 200 and 350 ms) does not provide this detection limit. However, the sensitivity can be improved by averaging the measurements for longer times, over which the instrument is stable. The stability of the IBBCEAS system is mainly affected by temperature fluctuations, mechanical instabilities and pressure drifts. In order to characterize the long-term stability of the instrument, two different studies were conducted on the IBBCEAS-$NO_2$ during the Antarctica field campaign, Dôme C 2019-2020 (the same results for the IBBCEAS-$NO_x$ can be found in the SI). For both studies, the light transmitted through the cavity ($I$) was integrated at the CCD for 250 ms, providing a signal-to-noise ratio of 110 for a single spectrum. The reference spectrum ($I_0$) was taken by averaging 2,000 individual spectra ($\sim$ 8 min) while flushing the cavity with zero-air. Subsequently, a 9h-long time series was recorded for each instrument maintaining the zero-air flow. The instrument was regulated at $12.0 \pm 0.2\,°C$, with a cavity pressure of $630.0 \pm 0.7$ mbar, and a gas flow of $1.02 \pm 0.11$ L $min^{-1}$. The minimum absorption coefficient ($\alpha_{min}$), corresponding to



the standard deviation of the residual of the spectrum, was deduced for different time averages. The results are shown in the

220 log-log plot of Figure 5, were the dots are the data and the dashed line indicates the trend in case of pure white noise regime. From the graph one can see that the instrument follows the white noise trend for about 22 min (5,200 averages), afterwards, the baseline noise start to deviate due to the arise of frequency dependent noise. The choosen $\alpha_{min}$ value corresponds to $1.8 \times 10^{-10}$ $cm^{-1}$ within $\sim 22$ min (5,200 spectra) of measurement during wich a reference spectrum in absence of absorbers and the absorption spectrum are acquire. The corresponding figure of merit (Noise Equivalent Absorption Sensitivity, NEAS or

225 $\alpha_{min}(BW) = \alpha_{min}$ x $\sqrt{\frac{t_{int}}{M}}$) is therefore $7.5 \times 10^{-11}$ $cm^{-1}$ $Hz^{-1/2}$ per spectral element (with $t_{int}$ the integration time, $\sim 11$ min, and M the number of independent spectral elements, here 800 spectral elements are considered for the spectral fit).

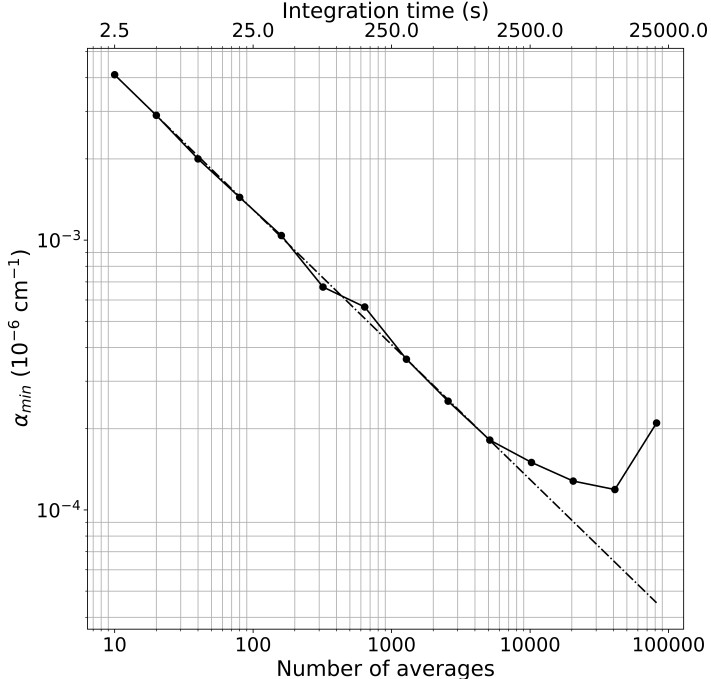

**Figure 5. The minimum absorption coefficient $\alpha_{min}$ versus the number of spectral average for the IBBCEAS-NO$_2$ instrument. For these measurements the cell was continuously flushed with a flow of 1.02 L min$^{-1}$ of zero-air, and the $\alpha_{min}$ was calculated from the standard deviation of the residual of the spectra at different time averages.**

For the same field time series an Allan-Werle (AW) statistical method on the measured concentrations was employed (Werle et al., 1993). In this case, spectra were averaged in block of ten and analysed by the fit routine. The results of the fit are reported on the top graph of Figure 6. For an acquisition time of 2.5 s, corresponding to 10 averaged spectra, the AW standard deviation

230 $\sigma_{AW-SD}$ was 230, 6.7, 195 ppt and 800 ppb for NO$_2$, IO , CHOCHO and O$_3$, respectively. By increasing the integration time, the $\sigma_{AW-SD}$ decreased following the white noise trend (colored dashed line of Figure 6 bottom) with a characteristic $\sqrt{N}$ slope (where $N$ is the number of averaged spectra). Because a reference spectrum in absence of absorbents is required by this



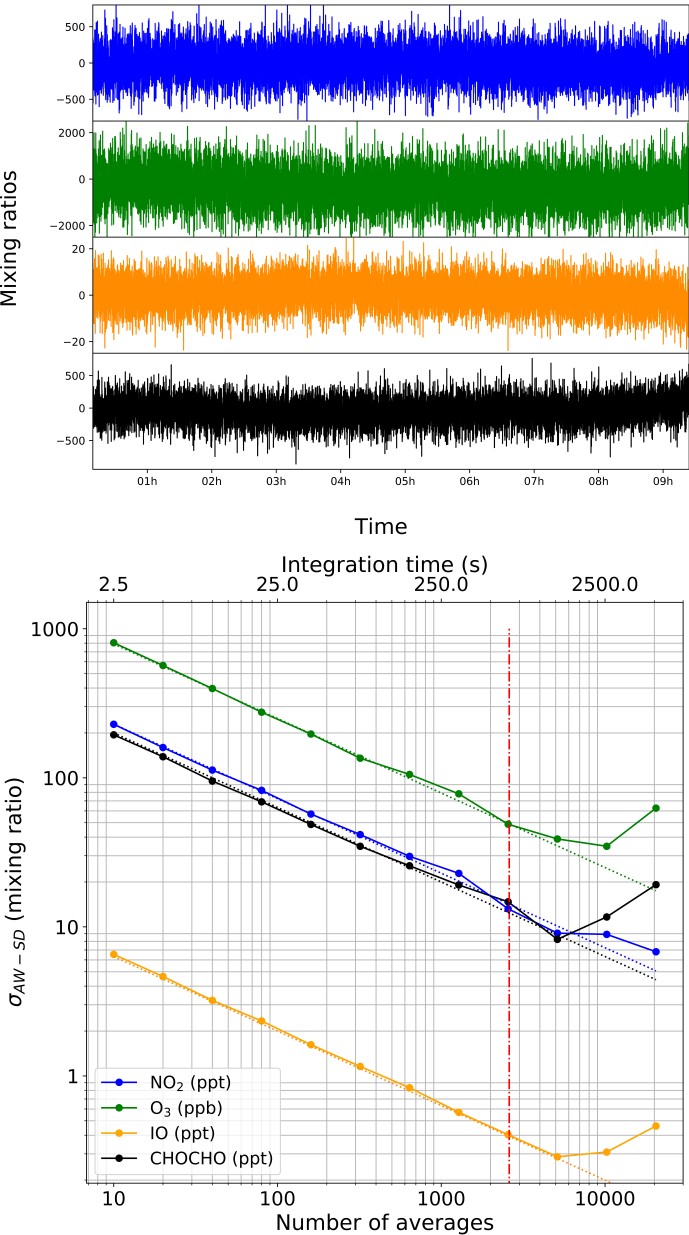

**Figure 6. (top) Mixing ratios of the target species $NO_2$, $O_3$, IO, and CHOCHO measured during a nine hours Allan-Werle variance statistical experiment flowing zero-air thought the cavity on the IBBCEAS-$NO_2$ instrument. (bottom) The log-log Allan-Werle standard deviation plot, illustrating that the instrument performance follow the white noise regime up to a certain extend, identified by the dashed lines. This represents the optimum integration time, after which instrumental instabilities start to dominate.**



CEAS technique, depending on the shape of the AW plot different strategies may be followed. In our case, the AW trends continue to decrease for all species during $\sim 22$ min (5,200 averages), this means that one can spend 11 minutes acquiring the reference spectrum and further 11 minutes for the absorption spectrum, leading to limits of detections (LODs) of 9, 0.3, 8 ppt and 40 ppb ($1\sigma$) for $NO_2$, IO , CHOCHO and $O_3$, respectively. In our case, we chose to divide the measurement times by two (i.e $\sim 11$ min and 2,600 averages for acquiring both the reference and the absorption spectra), offering equally interesting LODs: 15, 0.4, 16 ppt and 50 ppb for $NO_2$, IO, CHOCHO, and $O_3$ ($1\sigma$), respectively, and allowing us to stay within the white noise regime.

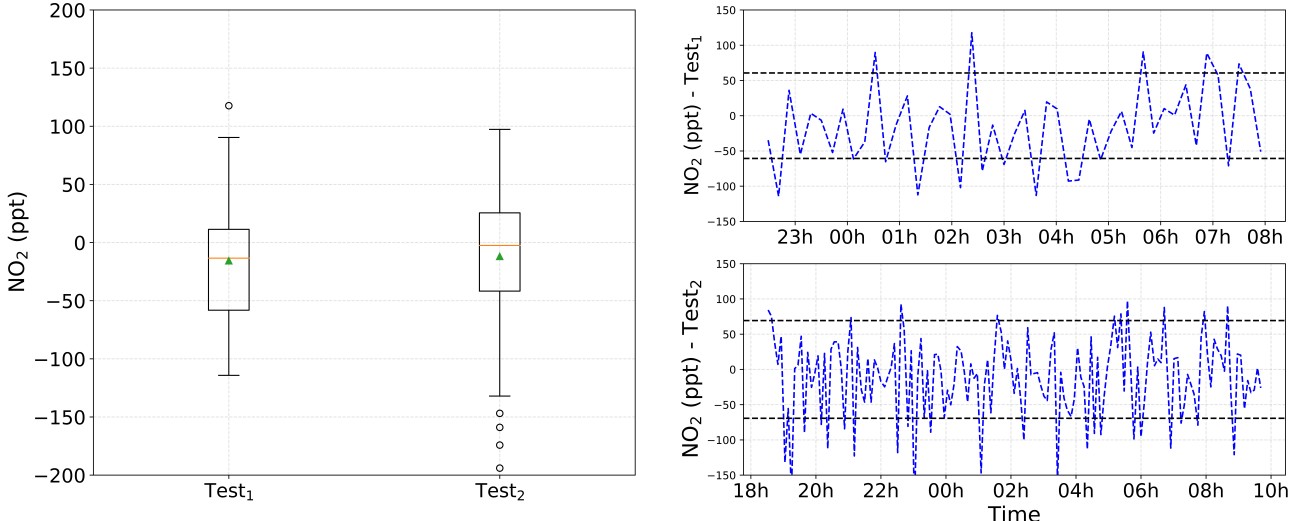

**Figure 7. (left) Boxplot of the stability test while continuously flowing zero-air in the cavity, means over 46 and 148 measurements (for Test 1 and Test 2, respectively), are shown as green triangles, dots represent the outliers. (right) Timeseries of two long-term stability tests. The results obtained by alternating reference and absorption spectra within a time interval corresponding to a number of averages where the instrument follows the white noise confirms the validity of the measurement strategy.**

Long-term stability of the instrument was further studied by taking regular reference spectra within the optimum white noise time of the instrument while continuously flushing the instrument with zero-air mixture. In this case $\sim 5$ min and $\sim 3$ min intervals were chosen, corresponding to 1000 and 580 averages (for 300 ms integration time) and a precision on the $NO_2$ concentrations of $\sim 20$ and $\sim 23$ ppt ($1\sigma$), respectively. The results are reported in figure 7. Test 1 (1000 averages) was run for 9h and test 2 (580 averages) for 15h. These tests highlights the reliability of the measurement protocol, with the long term measurements well distributed within the $3\sigma$ of the measurement precision (60 and 70 ppt respectively). A Box-plot is also reported representing the average values (green triangles) and medians, quartiles, minimum and maximum values. Table 1 hereafter shows a comparison between our instrument presented in this work and other recently developed IBBCEAS systems.

**Table 1. Comparisons of the performances with other IBBCEAS systems**





| References | Centered wavelengh (nm) | FWHM spectral resolution (nm) | NO$_2$ detection limit (1$\sigma$) | Integration time (s) | Cavity lengh (cm) | Mirror reflectivity (%) | Optical path lengh (km) |
|---|---|---|---|---|---|---|---|
| Jordan et al. (2019) | 505 | 30 | 124 ppt | 60 | 102 | 99.98 | $\sim 5$ |
| Min et al. (2016) | 455 | 18 | 40 ppt | 5 | 48 | 99.9973 | $\sim 18$ |
| Liu et al. (2019) | 455 | 18 | 9 ppt | 100 | 84 | > 99.98 | 4.2 |
| Liang et al. (2019) | 448 | 15 | $\sim 15$ ppt | 30 | 70 | > 99.995 | 14 |
| This work (2020) | 450 | 19 | 9 ppt | 650 | 41.7 | 99.9913 | $\sim 4.8$ |

## 4.4 Indirect measurement of NO

Measuring NO and NO$_2$ simultaneously is important to study the NO$_x$ budget in the atmosphere. In the selected blue visible region, there are no NO absorption features for direct optical measurements, and optical absorption detection of NO is typically done in the infrared region (Richard et al., 2018). However, its detection can be performed by indirectly measuring NO$_2$ after chemical conversion of NO to NO$_2$ in a controlled O$_3$ excess environment. This will lead to the measurement of NO$_x$, which, coupled by a simultaneous detection of NO$_2$ will provide the concentration of NO ([NO] = [NO$_x$] - [NO$_2$]) (Fuchs et al., 2009):

$$NO + O_3 \rightarrow NO_2 + O_2 \qquad k_1 = 1.80 \text{ x } 10^{-14} \text{ cm}^3 \text{ molec}^{-1} \text{ s}^{-1} \text{ at 25 °C} \qquad (R1)$$

O$_3$ was produced by electrolysis of water using commercial ozone-micro-cells (INNOVATEC) allowing the generation of O$_3$ without nitrogen oxides impurities and without the need of an oxygen gas bottle. The cells were mounted in a home-made plastic container offering a 200 cm$^3$ water reservoir. With a miniaturized design (15 x 15 x 15 cm$^3$), ozone production can be controlled upon injection into the inlet line. The sample air flow to be analyzed works as carrier gas for flushing the ozone enriched surface water. This design also prevents the production of unwanted oxidizing agents such as peroxides, as well as sample dilution, causing a signal degradation and requiring precise flow measurements for quantitative analysis. The production of O$_3$ is controllable by the amount of electrolytic cells used and the supplied current, offering a dynamic range of 0 – 50 ppm of O$_3$ for a 1 L min$^{-1}$ total flowrate. A diagram and details of the system can be found in Figure SI-5 and section 3.2 of the supplementary informations. For long-term use of the instrument, the overall water consumption should be considered. Losses due to evaporation were estimated to be between 7 and 30 cm$^3$ per day at 10 and 30 °C respectively for a flow rate of 1 L min$^{-1}$ while losses due to electrolysis are negligible, with only 0.024 cm$^3$ per day of consumption. The other parameter to consider is the mixing time between the ozone generator and the measurement cell with respect to the O$_3$ excess. For instance, the calculated production rate of NO$_2$ from (R1) (i.e. reaction speed or conversion rate of NO) is v = $4.20 \times 10^{11}$ molecules cm$^3$ s$^{-1}$ for 5 ppb of NO and 8 ppm of O$_3$. Under these conditions, a mixing time of 0.29 s is required for completing the conversion. With an air flow of 1 L min$^{-1}$, a 40-cm long 4-mm internal diameter tube is therefore required between the ozone generator and the measurement cell. The performance of the ozone generation system was tested on the IBBCEAS instrument with a nitrogen oxide standard gas bottle containing $\sim$ 180 ppb of NO in air (Air Liquide). Kinetic simulations using Tenua software were made in order to establish the O$_3$ excess concentrations needed to achieve the complete conversion of NO to





NO$_2$, which, along with its detection, was tested with the IBBCEAS instrument by varying the excess concentration of O$_3$ until complete conversion of NO was achieved at different flows (i.e. different reaction times before reaching the measurement cell). The experimental results were in good agreement with the simulations as reported in Figure SI-6. In addition, the instrument was found to have a linear response regarding the detection of the produced O$_3$. The detection limit for the NO$_x$ measurement

was found to be similar to the one of NO$_2$ (12 ppt (1$\sigma$) in 22 min of integration time) while for NO, retrieved as the difference between the NO$_x$ and the NO$_2$ concentrations, the detection limit estimated from the error propagation corresponds to 21 ppt.

## 5    Possible chemical and spectral interferences

Further possible interferences on NO$_2$ detection in the presence of high levels of O$_3$ were also studied, since a large excess of O$_3$ could trigger the following reactions with rate constants that are few orders of magnitude lower than k$_1$ (from the NIST

Kinetics Database):

NO$_2$ + O$_3$ → NO$_3$ + O$_2$                                                  k$_2$ = 3.8 x 10$^{-17}$ cm$^3$ molec$^{-1}$ s$^{-1}$ at 25 °C                 (R2)

NO$_2$ + O$_3$ → 2O$_2$ + NO                                                  k$_3$ = 1.0 x 10$^{-18}$ cm$^3$ molec$^{-1}$ s$^{-1}$ at 25 °C                 (R3)

NO$_3$ + O$_3$ → 2O$_2$ + NO$_2$                                            k$_4$ = 1.0 x 10$^{-17}$ cm$^3$ molec$^{-1}$ s$^{-1}$ at 25 °C                 (R4)

To study those possible interferences, 100 ppb of NO$_2$ produced by a permeation tube were pumped through the ozonizer and the spectrometer at a flow rate of 1 L min$^{-1}$ while varying the concentration of O$_3$ from 0 to 10 ppm. NO$_2$ concentration was stable at low ozone concentrations, while a drop of 14 % was observed at high levels of O$_3$ ($\geqslant$ 8 ppm). Kinetics simulations showed that the NO$_2$ consumption in favor of the NO$_3$ production (NO$_2$ + O$_3$ → NO$_3$ + O$_2$) was kinetically possible under

those conditions. The consumption of NO$_2$ is strongly dependent on the reaction time and the concentration of O$_3$. The later should be selected according to the reaction time imposed by the volume of the inlet line and the flow rate, therefore making this interference negligible. Other chemicals reactions could led to an overestimation of NO$_2$ mixing ratios:

HONO → NO + HO                                                            k$_5$ = 3.9 x 10$^{-21}$ cm$^3$ molec$^{-1}$ s$^{-1}$ at 25 °C                 (R5)

HO$_2$NO$_2$ → NO$_2$ + HO$_2$                                          k$_6$ = 1.3 x 10$^{-20}$ cm$^3$ molec$^{-1}$ s$^{-1}$ at 25 °C                 (R6)

Couach et al. estimated the background levels of HONO and HO$_2$NO$_2$ in Grenoble to be 4 and 2 ppq (or 10$^{-15}$ mol mol$^{-1}$), respectively (Couach et al., 2002). With such low concentrations and kinetic constant rates, interferences due to reactions (R5) and (R6) can be neglected in an urban envirionment. However, in remote areas such as the East Antarctic Plateau, HO$_2$NO$_2$ levels were estimated by indirect measurements to be around 25 ppt (Legrand et al., 2014). Because the lifetime of HO$_2$NO$_2$

decreases with temperature ($\tau_{\mathrm{HO_2NO_2}}$ = 8.6 h at -30°C and 645 mbar), its measurement using an instrument stabilized at higher temperature would lead to an overestimation of the NO$_2$ due to the thermal degradation of the HO$_2$NO$_2$. However, this interference can be minimized by working at low temperatures : at 10 °C and 1 L min$^{-1}$ flow in our IBBCEAS instrument, the NO$_2$ signal would be overestimated by only 1 ppt, which is below the detection limit of the sensor.The instruments were





therefore designed for working at low temperature (up to few degrees Celsius). Last reaction, (R7), may also lead to possible
interferences on the $NO_2$ detection:

$$HONO + OH \rightarrow NO_2 + H_2O \qquad\qquad k_7 = 4.89 \text{ x } 10^{-12} \text{ cm}^3 \text{ molec}^{-1} \text{ s}^{-1} \text{ at 25 °C} \qquad (R7)$$

In urban environments and remote regions, one can observed up to 4 x $10^6$ OH radicals cm$^{-3}$ (Heard, 2004; Mauldin et al.,
2001). With background levels of HONO such as 4 ppq in the city of Grenoble and around 30 ppt in Dôme C, Antarctica
(Legrand et al., 2014), very low mixing ratios of $NO_2$ (< few ppq) would be produced by (R7) in less than 8 s (residence time
of the molecules in the instrument at 1 L min$^{-1}$). Therefore, contribution from this interference can be neglected. Previous
works also highlighted possible artifacts through the heterogenous reaction of $NO_2$ and $H_2O$ occurring in thin films on surfaces:
the approximate rate production of HONO plus NO calculated in their study was reported to be between 4 x $10^{-2}$ and 8 x $10^{-2}$
ppb min$^{-1}$ per ppm of $NO_2$ (Finlayson-Pitts et al., 2003). Assuming linearity between production rates and concentrations,
this would represent a range of 8 to 16 ppq for 200 ppt of $NO_2$ in remote area such as the East Antarctic Plateau. The losses
that may occur on the thin films on surfaces through the heterogeneous reaction of $NO_2$ and $H_2O$ are therefore negligible.
Finally, detection of $NO_2$, CHOCHO and IO may be affected by spectral interferences. For instance, water vapour also shows
an absorption signature at this wavelength region which was included in the fit routine. Its spectral fit is important particularly
for the measurement of $NO_x$, where the inlet sampling line gets saturated in water vapor while passing through the water
reservoir of the ozone generator. In addition, artifacts on the signal and the spectral fit were studied by varying the $O_3$, $NO_2$
or NO mixing ratios in cavity. Small imperfections of the fit could lead to large effects on the $NO_2$ retrieved mixing ratio,
particularly at sub-ppb concentrations and in presence of large amounts of ozone. However, no appreciable effects of possible
artifacts were observed while $O_3$ concentrations up to 8 ppm were used. These performance studies and the simplicity of the
ozone generator, compact and fully controllable, make it suitable for field applications.

## 6    Conclusions

A compact, robust, affordable and highly sensitive IBBCEAS instrument for direct detection of $NO_2$, IO, CHOCHO and $O_3$
and indirect detection of NO is reported in this work. The instrument relies on the injection of incoherent radiation from
a compact, high power and low cost LED source, into a high-finesse optical cavity. The instrument provides a minimum
detectable absorption of $1.8 \times 10^{-10}$ cm$^{-1}$ corresponding to a figure of merit (Noise Equivalent Absorption Sensitivity, NEAS)
of $7.5 \times 10^{-11}$ cm$^{-1}$ Hz$^{-1/2}$ per spectral element. Thanks to the broadband feature, multi-species detection can be performed
with detection limits of 9 ($NO_2$), 0.3 (IO), 8 (CHOCHO) ppt and 40 ppb ($O_3$), $1\sigma$, within 22 minutes of measurements (which
account for the reference and absorption spectra acquisition). Detection limits for the indirect measurement of $NO_x$ and NO
are 12 and 21 ppt ($1\sigma$), respectively. The instrument has been designed to fit in a 19", 3U rack-mount case, weights 15 kg and
has a total electrical power consumption < 300 W. The detection limits could be further improved by replacing the ULS2048L
Avantes spectrometer, which offers at this working wavelength a signal to noise ratio on a single acquisition of 110 and a
sensitivity of 172,000 counts $\mu$W$^{-1}$ ms$^{-1}$, with a spectrometer with an integrated cooled CCD. The cooling would allow to



gain up to a factor of ten on the signal to noise ratio, which would directly apply to the detection limits. A better sensitivity of the CCD would also allow the use of higher reflectivity mirrors as done by Min et al. (Min et al., 2016) providing an effective optical path length of 18 km (with similar cavity length), $\sim$ 3 times higher than the one obtained in this work.

Its dynamic ranges, detection limits and multi-species detection character make this instrument well suitable for measurements
in different environments, from highly polluted to very remote areas such as polar regions. The instruments can be used in the future to address different scientific questions, related to the oxidative capacity at particular regions (i.e. inland and coastal polar atmospheres), where variability of $NO_x$ and IO would provide key information for understanding the mechanisms taking place in such remote areas. The detection of the $\alpha$-dicarbonyl CHOCHO may have applications at the marine boundary layer, where its source remains unknown and its contribution to secondary aerosol particle formation may be relevant (Ervens et al.,
2011; Volkamer et al., 2007; Fu et al., 2008).

*Data availability.* Available on request

*Author contributions.* The IBBCEAS instruments were designed and developed by CB under the supervision of RG. AB developed and validated the ozone generation. The instruments were optimized and validated by CB and AB who also did the instrumental intercomparison and the measurements for the long-term stability. RG was the principal supervisor of the project. JS and AD contributed with their knowledges
in atmospheric sciences and they closely followed the project with regular meetings. JS and RG are the supervisors of AB PhD thesis under which the instruments are deployed. NC provided technical and engineering inputs particularly at the beginning of the project. The manuscript was written by AB, CB and RG, with all authors contributions.

*Competing interests.* no competing interests are present

*Acknowledgements.* The research leading to these results has received funding from: PARCS project (Pollution in the ARCtic System,
a project of the CNRS French Artic Initiative: http://www.chantier-arctique.fr/en) ; the LabEx OSUG@2020 ("Investissements d'avenir" – ANR10 LABX56) ; the French National program LEFE (Les Enveloppes Fluides et l'Environnement) ; the Agence Nationale de la Recherche (ANR) via contract ANR-16-CE01-0011-01 EAIIST ; the Foundation BNP-Paribas through its Climate initiative program and by the French Polar Institute (IPEV) through programs 1177 (CAPOXI 35–75) and 1169 (EAIIST). The authors would like to thank the LabEx OSUG@2020 for funding the ThermoFisher™, 42i $NO_x$ analyzer and IPEV for funding the ThermoFisher 42iTL™, $NO_x$ trace analyzer that
were used during the development of the IBBCEAS instruments presented here. The authors thank G. Méjean and D. Romanini from LIPhy, France, for the very useful exchange of informations on IBBCEAS technique as well as A. A. Ruth from the University of Cork, Ireland, for his very useful feedbacks on the manuscript. And finally, the authors greatly thank the technical staff of the IGE for their technical support.



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
