# Peer review of "A compact Incoherent Broadband Cavity Enhanced Absorption Spectrometer (IBBCEAS) for trace detection of nitrogen oxides, iodine oxide and glyoxal at sub-ppb levels for field application"

_Atmospheric Measurement Techniques, 2020_

## Referee Comment (RC1) · Anonymous Referee #1 · 6 May 2020

Barbero et al. describe a cavity enhanced spectrometer for detection of NO2, IO, glyoxal, and O3. The instrument fits in a 19" rack mount and is temperature and pressure controlled and detection limits of 9, 0.3, 8 ppt and 40 ppb respectively in 22 minutes. The size and detection limits represent a step forward in the portability of IBBCEAS instruments, as well as the low power requirements. I recommend the paper for publication after addressing the following major and minor comments.

[Figure]

Major Comments:

Line 150: The authors present a calibration procedure using NO2 as the calibration gas calibrated against a chemiluminescence (CLD) NOx detector as the standard. This seems counter intuitive to use an instrument that has a multitude of known flaws with regards to NO2 detection and interference to standardize your instrument. If the authors had shown using a more consistent and reliable technique (such as the one employed by Washenfelder as referenced in the paper) and compared the mirror calibration to show that their NO2 process is reliable, then this would have been an acceptable way to proceed.

Figure 3: The authors then show a time trace of good agreement of the IBBCEAS instrument with a CLD instrument. Of course, this isn't surprising, since the IBBCEAS was calibrated to the CLD instrument.

Line 120: The argument that flow calculations show that the air doesn't impact the mirrors and therefore, no purge is necessary seems insufficient. Some air will impact on the mirrors, bringing humidity, organics and other material that will deposit out on the surface of the mirror and degrade the reflectivity over time. The authors present no further justification for whether this worked. What was the rate of decay in the mirror reflectivity over time? Did the lower reflectivity to start with impact the ability to get away with this set up?

Line 212: Here it states that the instrument is sensitive to temperature and pressure drifts. While these all together can be tested through the variance analysis presented (in combination with any drift in the spectrometer noise), was there any effort to quantify how sensitive the instrument is to pressure changes?

Table 1: Comparisons are made to other IBBCEAS systems. While this is good, there is no effort to show them in a head to head comparison with comparable integration times which seems less useful, especially as the integration time listed for this instrument is 6 times longer than the next longest time in the table.

[Figure]

Line 351: This appears to be in conflict with the journal data policy. The data must be available in a repository or other source, not just on request.

Minor Comments:

Title: glyoxal is listed as a species of interest but never demonstrated. O3 while demonstrated is only useful for the NOx (NO+NO2) version of the instrument in verifying how much O3 is being used to titrate the NO. 40 ppb is not a useful LOD for ambient O3 measurement.

Line 56: "Leading (to) different"

Section 4.2: It would be simple to use the IBBCEAS instrument as the primary standard for the NO2 determination for the bottle if calibrated with N2 and He as described previously in the literature. Given the issues with CLD instruments and how extremely far off the measured bottle concentration was from the standard.

Figure 6 caption: "Certain extend" change to extent.

Figure 7 caption: How important are the outliers? They seem to be very far out. Is there something that caused them that they could be filtered out and removed in the analysis. It would be reasonable to remove 10 points out of >5000 if there was some software or hardware issue (pressure spike) that caused them.

Table 1: The column labeled FWHM is not the instrument resolution, but the fit window, update to be consistent (if the FWHM was 30 nm, the instrument would not be measuring any of these species).

Line 274: Provide a reference for the Tenua software.

Line 309: Replace "Last reaction" with "One more reaction"

Line 312: Change to "In urban environments OH radicals can be observed up to 4 x 10ˆ6 cmˆ-3"

Line 322: Mention is made here with regard to water interference, and that it is fit, but no accuracy is stated for the retrieved water concentrations or their effect on the fits of other species and the RMS noise.

Line 333: "absorption", this should be extinction. IBBCEAS instruments measure the sum of absorption + scattering (extinction).

Line 334: "Thanks to the broadband feature", the broadband feature or features of which species? Usually, these fits are sensitive to the narrow-band features which is what allows for simultaneous detection of multiple species.

Line 341: "A better", just start with Better...

Line 344: Revise to "The dynamic range, detection limits, and"

---

## Referee Comment (RC2) · Anonymous Referee #2 · 10 May 2020

The manuscript "A compact Incoherent Broadband Cavity Enhanced Absorption Spectrometer (IBBCEAS) for trace detection of nitrogen oxides, iodine oxide and glyoxal at sub-ppb levels for field application" by Barbero et al. reports a LED-based IBBCEAS for the measurement of $NO_2$, IO, CHOCHO, and $O_3$. By coupling an $O_3$ generator, concentrations of NO in ambient air can also be measured by the IBBCEAS. The volume and weight of the instrument are suitable for field application. Overall, the manuscript presents valuable information on the IBBCEAS application which deserves publishing on AMT. However, I do have the following comments which I would like to the authors to be addressed before the final publication.

**General Comments:**

1. As shown in Table 1 in the manuscript, compared with the reported IBBCEAS whose wavelength centered around 450 nm, the instrument introduced here is inferior in terms of mirror reflectivity, optical path length, and time resolution. If the IBBCEAS introduced here cannot be improved from the aspects of above key parameters, novelty of this work should be detailed and highlighted. In addition, authors need to carefully check the data listed in Table 1, the reflectivity and optical path length of Liu et al.'s IBBCEAS is 0.99993 and 10.3 km, respectively (Liu et al., 2019).

2. The description of measuring CHOCHO in the manuscript is limited, as the Fig. 2(b) only showed simultaneously detection of $NO_2$, IO, and $O_3$. It should be better if the authors could present a graph which contains 5 gas absorbers ($NO_2$, IO, $O_3$, CHOCHO, and $H_2O$) simultaneous retrieving. It should be noted that the concentrations of $NO_2$, $O_3$ shown in Fig. 2(b) were significantly higher than their concentrations in ambient air, even in polluted area. As the purpose of the manuscript is to present an instrument for field application, it would be more persuasive for readers if a fitting example with low concentrations of gas absorbers could be provided.

3. The manuscript does not provide information about uncertainty of the instrumental

measurements, As to the limit of detection (LOD), authors seems to confuse the concepts among LOD, sensitivity, and precision, because these three words appear alternately in Sect. 4.3.1. The using of these concepts needs to be clarified and revised in the manuscript.

**Specific Comments:**

Line 56ff: References should not be quoted twice in the same sentence if it is already be written at the beginning. For example, "Venables et al. (2006) were …… (Venables et al., 2006).".

Line 64: "Min et al 2016" -> "Min et al. (2016)"

Line 65: "very high reflective mirrors […]"

Line 130: Eq. (1) There are probably better ways to format the equations such that the size of the brackets is matched to the size of the arguments within the bracket.

Line 150: "Washenfelder et al. (2008) described […]"

Line 152: "(e.g., helium versus air or nitrogen) […]"

Line 153: Such an approach to calculate mirror reflectivity has been proposed before (Venables et al., 2006) and has been used by previous studies (e.g., Duan et al., 2018). It would be better to reorganized the sentences in another way in the manuscript. In addition, did authors compare the difference between two reflectivity calibration methods based on their own IBBCEAS?

Page 8, Figure 2 (a): The text-label (i.e., Reflectivity) on the y-axis was covered.

Line 202: In addition to the discrepancies at low $NO_2$ concentrations, obvious discrepancies measured by two instruments can also be observed at high $NO_2$ conditions, e.g., 18/10/01 - 09:00 and 19/07/19 – 05:45. Could authors provide an explanation about the phenomenon?

Page 13, Figure 6 (top): The units of mixing ratio was missing.

Page 14, Figure 7: The left Box-plot is not as useful as drawing a histogram which contains measured NO2 concentrations when performing empty cavity measurements. Such a histogram can not only be used to show averages, but also be used to estimate LOD from the frequency number of histogram distribution.

Line 247: A short discussion about the comparison shown in Table 1 is better than only presenting a Table without any explanation.

Line 308: "[…] sensor.The instruments […]" -> "[…] sensor. The instruments […]"

Line 324: As the inlet sampling line gets saturated in water vapor while passing through the ozone generator, did authors quantify the influence on CHOCHO measurements? For example, measure the CHOCHO standards with and without using ozone generator.

Reference:

Duan, J., Qin, M., Ouyang, B., Fang, W., Li, X., Lu, K., Tang, K., Liang, S., Meng, F., Hu, Z., Xie, P., Liu, W., and Häsler, R.: Development of an incoherent broadband cavity-enhanced absorption spectrometer for in situ measurements of HONO and NO2 Atmos Meas Tech, 11, 4531-4543, 2018.

Liu, J., Li, X., Yang, Y., Wang, H., Wu, Y., Lu, X., Chen, M., Hu, J., Fan, X., Zeng, L., and Zhang, Y.: An IBBCEAS system for atmospheric measurements of glyoxal and methylglyoxal in the presence of high NO2 concentrations, Atmos Meas Tech, 12, 4439-4453, 2019.

Venables, D. S., Gherman, T., Orphal, J., Wenger, J. C., and Ruth, A. A.: High Sensitivity in Situ Monitoring of NO3 in an Atmospheric Simulation Chamber Using Incoherent Broadband Cavity-Enhanced Absorption Spectroscopy, Environ Sci Technol, 40, 6758-6763, 2006.

---

## Author Comment (AC1) · 9 Jul 2020

Dear Editors, We express our gratitude for the time and effort dedicated to the reviewing of our submitted manuscript. We worked diligently to address all the concerns raised by the referees and we thank the reviewers for the pertinent remarks that allowed to improve the manuscript. Attached we provide our detailed response to their comments. We hope that the applied revisions are to the satisfaction of the editors.

[Figure]

Kind regards, Albane Barbero, Camille Blouzon, Joël̀Ĺl Savarino, Nicolas Caillon, Aurel̀Ą̧lien Dommergue, and Roberto Grilli

Please also note the supplement to this comment:
https://www.atmos-meas-tech-discuss.net/amt-2020-104/amt-2020-104-AC1-supplement.pdf

─────────────────────────

[Figure]

**Supplement:**

**A compact Incoherent Broadband Cavity Enhanced Absorption Spectrometer (IBBCEAS) for trace detection of nitrogen oxides, iodine oxide and glyoxal at sub-ppb levels for field application**

**Revision Notes**

Albane Barbero[1], Camille Blouzon[1], Joël Savarino[1], Nicolas Caillon[1], Aurélien Dommergue[1], and Roberto Grilli[1]

[1]Univ. Grenoble Alpes, CNRS, IRD, Grenoble INP*, IGE, 38000 Grenoble, France

*Institute of Engineering Univ. Grenoble Alpes

Correspondence: Roberto Grilli (roberto.grilli@cnrs.fr)

Dear Editors,

We express our gratitude for the time and effort dedicated to the reviewing of our submitted manuscript. We worked diligently to address all the concerns raised by the referees and we thank the reviewers for the pertinent remarks that allowed to improve the manuscript. Below we provide our detailed response to their comments. We hope that the applied revisions are to the satisfaction of the editors.

Kind regards,

Albane Barbero, Camille Blouzon, Joël Savarino, Nicolas Caillon,
Aurélien Dommergue, and Roberto Grilli

Manuscript information

https://doi.org/10.5194/amt-2020-104 Preprint.

**Title** "A compact Incoherent Broadband Cavity Enhanced Absorption Spectrometer (IBBCEAS) for trace detection of nitrogen oxides, iodine oxide and glyoxal at sub-ppb levels for field application"

**Authors** Albane Barbero, Camille Blouzon, Joël Savarino, Nicolas Caillon, Aurélien Dommergue, and Roberto Grilli

**Submitted to** Atmospheric Measurement techniques

**Reviewer 1:**

Comments I and II (Reviewer 1):

*Line 150: The authors present a calibration procedure using $NO_2$ as the calibration gas calibrated against a chemiluminescence (CLD) $NO_x$ detector as the standard. This seems counter intuitive to use an instrument that has a multitude of known flaws with regards to $NO_2$ detection and interference to standardize your instrument. If the authors had shown using a more consistent and reliable technique (such as the one employed by Washenfelder as referenced in the paper) and compared the mirror calibration to show that their $NO_2$ process is reliable, then this would have been an acceptable way to proceed.*

*Figure 3: The authors then show a time trace of good agreement of the IBBCEAS instrument with a CLD instrument. Of course, this isn't surprising, since the IBBCEAS was calibrated to the CLD instrument.*

Answer: The tight schedule between the development of the instruments and their first use in the field prevented a comparison experiment between the method described in the paper and the Rayleigh scattering method described by Washenfelder et al (2008) for determining the mirrors reflectivity. Fortunately, the instruments came back from the field early June and we were able to do the Rayleigh experiment using standard He gas (Messer, Helium 5.0, 99.999%) and standard $N_2$ gas (Air Liquide, Alpha Gaz 2, 99.9999 %) cylinders, 5 µm Whatman® filters and taking into account the CCD dark noise. In addition, in between the field expeditions and the return of the instruments, we received a calibrator (Gas Standard Generator FlexStream™, Kin-Tek Analytical, Inc.) able to produce a stable $NO_2$ source. The sample is produced using a permeation tube of $NO_2$ (Kin-Tek ELSRT2W) calibrated at an emission rate of 115 ng min$^{-1}$ at 40 °C loaded into the calibrator. This type of calibrator is ideally suited for creating trace concentration mixtures (from ppt to ppm). Despite those efforts, we were not satisfied by the results of this calibration method because of several arguments: one of them being the discrepancies between the Rayleigh cross sections provided by Min et al (2016) (empirical values) against the theoretical cross sections using equations provided by Thalman et al. (2014). The results of the experiment are shown in the Figure below : with Rayleigh curve we obtain 74.6 ppb of $NO_2$ using Min cross sections and 98.0 ppb of $NO_2$ using Thalman cross sections, while the Kin-Tek $NO_2$ source was set at 49.6 ppb. The retrieved curve for matching $NO_2$ absorption cross sections is the blue one at the top, which also better match in shape the expected theoretical curve provided by the manufacturer and which provide the best residue ($1.0E^{-8}$ cm$^{-1}$ against 1.3 $E^{-8}$ and $1.8E^{-8}$ cm$^{-1}$ with Min and Thalman cross-sections, respectively). To confirm that the shape was correct, we further compared the convoluted literature absorption cross sections of CHOCHO with the experimental data (which applies our experimental reflectivity curve) and we obtain a good matching, confirming that the shape of the curve is correct (see Fig SI – 3).

[Figure]

We were able to do a calibration using the approach described in the manuscript, against the FlexStream™ calibrator delivering $NO_2$ at $49.6 \pm 0.2$ ppb, allowing us to identify any possible doubts regarding the $NO_2$ bottle used in the first instance. As shown in Figure 4(b) (reported here below), the instruments are well calibrated and have a linear response within a large range of concentrations.

[Figure]

**Comment III (Reviewer 1):**

*Line 120: The argument that flow calculations show that the air doesn't impact the mirrors and therefore, no purge is necessary seems insufficient. Some air will impact on the mirrors, bringing humidity, organics and other material that will deposit out on the surface of the mirror and degrade the reflectivity over time. The authors present no further justification for whether this worked. What was the rate of decay in the mirror reflectivity over time? Did the lower reflectivity to start with impact the ability to get away with this set up?*

Answer: The Method part in the supplementary has been modified to better explain our approach:

"1 Method

1.1 Solidworks simulation

[Figure]

**Figure SI - 1.** *SolidWorks simulations of the air flow entering the cavity at 1 L min$^{-1}$. (top) Turbulences created by the presence of a dead-volume between the cavity mirror and the gas exhaust. (bottom) Configuration with an optimum distance between the high reflective mirror and the gas outlet which maximize the effective optical pathlength, avoiding the use of purging gas at the mirrors while preserving the mirror cleanliness during the measurement. The gas inlet is placed at the center of the cavity.*

Solid works simulations were made with two different mirrors positions without purge flow (Fig SI - 1). With the cavity mirrors placed one or two centimeters away from the exit of the air flow, presence of turbulences in front of the mirrors can be observed, which may compromise the long term cleanliness of the high reflectivity mirrors, Fig SI – 1 (top). By placing the mirrors close to the gas flow exit, the absence of dead-volume minimizes the residence time of particles by avoiding localized turbulences to take place as shown in Fig SI – 1 (bottom) and prevents the mirrors surface from deposition of dust and organic matter. In addition, PTFE membrane filters (Whatman$^{®}$ PTFE membrane filters – TE 38, 5 μm, 47 mm) placed at the entrance of each sampling lines, reference and sample, prevent particles to enter the gas lines.

1.2 Mirrors cleanliness monitoring

A photodiode was mounted on a cap placed in front of the LED assemble. This allow to continuously monitor the LED intensity as $PD_{meas}$. After a calibration with a standard gas, the value at the photodiode and the mean light intensity at the CCD, while flushing with zero air and averaged over all the pixels, are stored as $PD_{calibr}$ and $I_{0-calibr}$, respectively. At any time, the expected intensity, $I_{0-expected}$, at the CCD can be calculated as follow:

$$I_{0-expected} = I_{0-calibr} \left( \frac{PD_{meas}}{PD_{calibr}} \right)$$

and compare to the intensity at the CCD during measurements, $I_{0-meas}$. The ratio $I_{0-meas}$ / $I_{0-expected}$ is therefore a direct indicator of the mirrors cleanliness as the variability of the LED intensity is accounted for in real time. Fig SI - 2 shows a timeseries of 10 days measurements during which no calibrations

were made. The variability of the LED intensity, Fig SI - 2(a), is less than 0.05 % over 10 continuous days, implying that the variability of the signal intensity, Fig SI - 2(b), only represents the mirror cleanliness over time. The variability of the latter, being less than 2 % (3 σ), validates the stability of the mirror reflectivity over time with the described set-up of our instruments and without purge flow.

[Figure]

***Figure SI – 2:*** *Monitored signals over 10 continuous days of measurement without any adjustment or calibration of the instrument. (a) $PD_{meas} / PD_{calibr}$ and (b) $I_{0-meas} / I_{0-expected}$.*"

Comment IV (Reviewer 1):

*Line 212: Here it states that the instrument is sensitive to temperature and pressure drifts. While these all together can be tested through the variance analysis presented (in combination with any drift in the spectrometer noise), was there any effort to quantify how sensitive the instrument is to pressure changes?*

Answer: The instrument being designed to measure in a remote environment such as the Antarctic plateau where the average pressure is 650 mbar, it was decided from the start to regulate the pressure inside the cavity, justifying why no measurement to quantify how sensitive the instrument is to pressure changes was made. However, some precisions were added

- to the manuscript line 107: "a pressure sensor P (STS ATM.ECO – accuracy ± 0.2 %)"
- to the supplementary as long with the following Fig SI – 7: "While the pressure is monitored at ± 0.2 % accuracy, the cavity's temperature is regulated to be 3 °C above ambient room temperature using a temperature controller and heat bands, as described in the manuscript. AW analysis were made with and without those heat bands to quantify how sensitive the instruments were to temperature changes. Fig SI – 7 shows the results obtained with the IBBCEAS-$NO_2$ instrument. Fig SI – 7 (left) shows the results with the heating bands turned off and one can see a deviation from the white noise after 1,300 averages with a maximum at ∼ 10,000 averages or ∼ 42 minutes corresponding to the laboratory temperature regulation cycle. However, by regulating the instrument temperature with the heating bands, the instrument is stable for longer

time, and is no longer affected by the external temperature variabilities as shown in Fig SI – 7 (right) ; similar results were observed for the IBBCEAS-NO$_x$ instrument."

[Figure]

**Comment V (Reviewer 1):**

*Table 1: Comparisons are made to other IBBCEAS systems. While this is good, there is no effort to show them in a head to head comparison with comparable integration times which seems less useful, especially as the integration time listed for this instrument is 6 times longer than the next longest time in the table.*

Answer: The table was carefully checked and modified. A short text was added to comment on the differences observed between the recently developed instruments:

"Table 1 shows a comparison between the instrument presented in this work and other recently developed IBBCEAS systems. The detection limits are given in ppt min$^{-1}$ (1$\sigma$) with the normalization time that accounts for the acquisition of the reference (without absorption) and sample spectra to allow a better comparison. It should be noticed that all the other developments took advantage from an optical spectrometer with a cooled CCD device to reduce dark noise. A more compact and affordable spectrometer was preferred in this work. The cooling at the CCD would allow to gain up to a factor of ten on the signal to noise ratio, which would directly apply to the achievable detection limits. Furthermore, a CCD with a higher sensitivity would allow to select higher reflective mirror and increase the optical pathlength. Noteworthy, the optimum integration time, corresponding to a minimum of the $\sigma_{AW-SD}$, is at 1,300 s ($\sim$ 22 min), allowing to achieve low detection limits even without a cooled CCD."

**Table 1. Comparisons of the performances with other recently developed IBBCEAS systems**

| References | Centered wavelengh (nm) | Source FWHM (nm) | NO$_2$ detection limit (ppt min$^{-1}$) | Sample path lengh (cm) | Mirror reflectivity (%) | Optical lengh (km) | Mirrors purged | CCD cooled (°C) | Minimum $\sigma_{AW-SD}$ deviation (s) |
|---|---|---|---|---|---|---|---|---|---|
| Min et al. (2016) | 455 | 18 | 16 | 48 | 99.9973 | 17.8 | no | -70 | 100 |
| Jordan et al. (2019) | 505 | 30 | 200 | 102 | 99.98 | 5.1 | yes | -80 | 300 |
| Liu et al. (2019) | 455 | 18 | 33 | 84 | 99.993 | 10.3 | yes | -70 | 100 |
| Liang et al. (2019) | 448 | 15 | 15 | 58.9 | 99.9942 | 11.7 | yes | -10 | 3,500 |
| This work (2020) | 450 | 19 | 40 | 41.7 | 99.9905 | 4.4 | no | no | 1,300 |

Comment VI (Reviewer 1):

*Line 351: This appears to be in conflict with the Journal data policy. The data must be available in a repository or other source, not just on request.*

Answer: "The data used in this study are available from the corresponding author upon request" is commonly used by other papers published in AMT. The folder would be ready to be inserted in a repository if the journal requires to do so.

Additional comments (Reviewer 1):

1. *Title: Glyoxal is listed as a species of interest but never demonstrated. $O_3$ while demonstrated is only useful for the $NO_x$ ($NO+NO_2$) version of the instrument in verifying how much $O_3$ is being used to titrate the NO. 40 ppb is not a useful LOD for ambient $O_3$ measurement.*

Answer: The reviewer is correct, the LOD for $O_3$ is indeed not enough for its monitoring at typical ambient concentrations. But this was not a goal of this development, and the $O_3$ monitoring turned out to be profitable for monitoring the NO to $NO_2$ conversion to assure the complete and only conversion of NO to get the $NO_x$ measurement. With the instruments being back from the field, we were able to demonstrate the measurements of Glyoxal (see Figure 2 modified in the manuscript).

2. *Line 56: Leading (to) different":* corrected

3. *Section 4.2: It would be simple to use the IBBCEAS instrument as the primary standard for the $NO_2$ determination for the bottle if calibrated with $N_2$ and He as described previously in the literature. Given the issues with CLD instruments and how extremely far off the measured bottle concentration was from the standard.*

Answer: The comment is already addressed in the answer for major Comments I and II.

4. *Figure 6 caption: "Certain extend" change to extent :* corrected.

5. *Figure 7 caption: How important are the outliers? They seem to be very far out. Is there something that caused them that they could be filtered out and removed in the analysis. It would be reasonable to remove 10 points out of >5000 if there was some software or hardware issue (pressure spike) that caused them.*

Answer: We did not find a reason for disregarding those outliers. Without a validated justification that would explain how the outliers came to be, it is preferred to leave them since they don't impact the conclusion of the experiment. However, it should be noticed that even those outliers falls within the 3σ uncertainty range, the Figure 7 has been adjusted to better highlight this observation.

6. *Table 1: The column labeled FWHM is not the instrument resolution, but the fit window, update to be consistent (if the FWHM was 30 nm, the instrument would not be measuring any of these species).*

Answer: We are sorry for the confusion, it was indeed the source width as the reviewer remarked. Regarding the table, as explained in the answer to Comment V, the table has been modified accordingly to the Reviewers comments.

7. *Line 274: Provide a reference for the Tenua software:* The following reference has been added: "Wachsstock, D.: Tenua: the kinetics simulator for Java; http://bililite.com/tenua., http://bililite.com/tenua, 2007."

8. *Line 309: Replace "Last reaction" with "One more reaction":* corrected.

9. *Line 312: Change to "In urban environments OH radicals can be observed up to 4 x $10^6$ $cm^{-3}$:* corrected.

10. *Line 322: Mention is made here with regard to water interference, and that it is fit, but no accuracy is stated for the retrieved water concentrations or their effect on the fits of other species and the RMS noise.*

Answer: Atmospheric measurements were done with and without fitting $H_2O$ to quantify the fitting interferences on $NO_2$. The Figure below shows the FIT results without (left), and with (right) $H_2O$ being included in the FIT routine. For this particular measurements, the results were giving 262.4 and 301.8 ppt of $NO_2$ and 4.7 and 4.4 ppm of $O_3$, respectively without and with the $H_2O$, leading, for this measurement, to an underestimation of 13 % on the $NO_2$ mixing ratio with the presence of 0.44 % humidity added by the $O_3$ production system in the sample line.

[Figure]

11. *Line 333: "absorption", this should be extinction. IBBCEAS instruments measure the sum of absorption + scattering (extinction).*

Answer: This remark is theoretically correct. The instrument does measure extinction because, even with a filter at the entrance of the spectrometer, we cannot completely remove particles. However, since the interest here lies on the molecular absorption measurement, we prefer to maintain this nomenclature "minimum detectably absorption coefficient" which will also be more consistent with the other literature works.

12. *Line 334: "Thanks to the broadband feature", the broadband feature or features of which species? Usually, these fits are sensitive to the narrow-band features which is what allows for simultaneous detection of multiple species:* The sentence has been modified to a better understanding.

13. *Line 341: "A better", just start with Better:* corrected.

14. *Line 344: Revise to "The dynamic range, detection limits, and":* corrected.

Additional remark (from the authors):

All the English mistake or the typo corrections suggested by the Reviewers have been corrected. Other changes coming from the opportunities of new experiments from the comments of the Reviewers were made. All the changes can be found in red in the manuscript and supplementary. Also, the following references where added or corrected:

− Duan, J., Qin, M., Ouyang, B., Fang, W., Li, X., Lu, K., Tang, K., Liang, S., Meng, F., Hu, Z., Xie, P., Liu, W., and Häsler, R.: Development of an incoherent broadband cavity-enhanced absorption spectrometer for in situ measurements of HONO and $NO_2$ Atmos Meas Tech, 11, 4531-4543, 2018.

− Liu, J., Li, X., Yang, Y., Wang, H., Wu, Y., Lu, X., Chen, M., Hu, J., Fan, X., Zeng, L., and Zhang, Y.: An IBBCEAS system for atmospheric measurements of glyoxal and methylglyoxal in the presence of high $NO_2$ concentrations, Atmos Meas Tech, 12, 4439-4453, 2019.

− Venables, D. S., Gherman, T., Orphal, J., Wenger, J. C., and Ruth, A. A.: High Sensitivity in Situ Monitoring of $NO_3$ in an Atmospheric Simulation Chamber Using Incoherent Broadband Cavity-Enhanced Absorption Spectroscopy, Environ Sci Technol, 40, 6758-6763, 2006.

− Villena, G., Bejan, I., Kurtenbach, R., Wiesen, P., and Kleffmann, J.: Interferences of commercial $NO_2$ instruments in the urban atmosphere and in a smog chamber, Atmospheric Measurement Techniques, 5, 149–159, https://doi.org/10.5194/amt-5-149-2012, https://www.atmos-meas-tech.net/5/149/2012/, 2012.

− Volkamer, R., Spietz, P., Burrows, J., and Platt, U.: High-resolution absorption cross-section of glyoxal in the UV–vis and IR spectral ranges, Journal of Photochemistry and Photobiology A: Chemistry, 172, 35–46, https://doi.org/10.1016/j.jphotochem.2004.11.011, https://linkinghub.elsevier.com/retrieve/pii/S1010603004005143, 2005.

− Wachsstock, D.: Tenua: the kinetics simulator for Java; http://bililite.com/tenua., http://bililite.com/tenua, 2007.

---

## Author Comment (AC2) · 9 Jul 2020

Dear Editors, We express our gratitude for the time and effort dedicated to the reviewing of our submitted manuscript. We worked diligently to address all the concerns raised by the referees and we thank the reviewers for the pertinent remarks that allowed to improve the manuscript. Attached we provide our detailed response to their comments. We hope that the applied revisions are to the satisfaction of the editors.

[Figure]

Kind regards, Albane Barbero, Camille Blouzon, JoeÌĹl Savarino, Nicolas Caillon, AureÌĄlien Dommergue, and Roberto Grilli

Please also note the supplement to this comment:
https://www.atmos-meas-tech-discuss.net/amt-2020-104/amt-2020-104-AC2-supplement.pdf

———————————————————

[Figure]

**Supplement:**

**A compact Incoherent Broadband Cavity Enhanced Absorption Spectrometer (IBBCEAS) for trace detection of nitrogen oxides, iodine oxide and glyoxal at sub-ppb levels for field application**

**Revision Notes**

Albane Barbero[1], Camille Blouzon[1], Joël Savarino[1], Nicolas Caillon[1], Aurélien Dommergue[1], and Roberto Grilli[1]

[1]Univ. Grenoble Alpes, CNRS, IRD, Grenoble INP*, IGE, 38000 Grenoble, France *Institute of Engineering Univ. Grenoble Alpes

Correspondence: Roberto Grilli (roberto.grilli@cnrs.fr)

Dear Editors,

We express our gratitude for the time and effort dedicated to the reviewing of our submitted manuscript. We worked diligently to address all the concerns raised by the referees and we thank the reviewers for the pertinent remarks that allowed to improve the manuscript. Below we provide our detailed response to their comments. We hope that the applied revisions are to the satisfaction of the editors.

Kind regards,

Albane Barbero, Camille Blouzon, Joël Savarino, Nicolas Caillon, Aurélien Dommergue, and Roberto Grilli

Manuscript information

https://doi.org/10.5194/amt-2020-104 Preprint.

**Title** "A compact Incoherent Broadband Cavity Enhanced Absorption Spectrometer (IBBCEAS) for trace detection of nitrogen oxides, iodine oxide and glyoxal at sub-ppb levels for field application"

**Authors** Albane Barbero, Camille Blouzon, Joël Savarino, Nicolas Caillon, Aurélien Dommergue, and Roberto Grilli

**Submitted to** Atmospheric Measurement techniques

**Reviewer 2:**

Comments I (Reviewer 2):

*As shown in Table 1 in the manuscript, compared with the reported IBBCEAS whose wavelength centered around 450 nm, the instrument introduced here is inferior in terms of mirror reflectivity, optical path length, and time resolution. If the IBBCEAS introduced here cannot be improved from the aspects of above key parameters, novelty of this work should be detailed and highlighted. In addition, authors need to carefully check the data listed in Table 1, the reflectivity and optical path length of Liu et al.'s IBBCEAS is 0.99993 and 10.3 km, respectively (Liu et al., 2019).*

Answer: The table was carefully checked and modified in order to highlight the novelty of this work on key parameters. A short text was added to comment on the differences observed between the recently developed instruments :

"Table 1 shows a comparison between the instrument presented in this work and other recently developed IBBCEAS systems. The detection limits are given in ppt min$^{-1}$ (1$\sigma$) with the normalization time that accounts for the acquisition of the reference (without absorption) and sample spectra to allow a better comparison. It should be noticed that all the other developments took advantage from an optical spectrometer with a cooled CCD device to reduce dark noise. A more compact and affordable spectrometer was preferred in this work. The cooling at the CCD would allow to gain up to a factor of ten on the signal to noise ratio, which would directly apply to the achievable detection limits. Furthermore, a CCD with a higher sensitivity would allow to select higher reflective mirror and increase the optical pathlength. Noteworthy, the optimum integration time, corresponding to a minimum of the $\sigma_{AW\text{-}SD}$, is at 1,300 s (~ 22 min), allowing to achieve low detection limits even without a cooled CCD."

**Table 1. Comparisons of the performances with other recently developed IBBCEAS systems**

| References | Centered wavelengh (nm) | Source FWHM (nm) | NO$_2$ detection limit (ppt min$^{-1}$) | Sample path lengh (cm) | Mirror reflectivity (%) | Optical lengh (km) | Mirrors purged | CCD cooled (°C) | Minimum $\sigma_{AW-SD}$ deviation (s) |
|---|---|---|---|---|---|---|---|---|---|
| Min et al. (2016) | 455 | 18 | 16 | 48 | 99.9973 | 17.8 | no | -70 | 100 |
| Jordan et al. (2019) | 505 | 30 | 200 | 102 | 99.98 | 5.1 | yes | -80 | 300 |
| Liu et al. (2019) | 455 | 18 | 33 | 84 | 99.993 | 10.3 | yes | -70 | 100 |
| Liang et al. (2019) | 448 | 15 | 15 | 58.9 | 99.9942 | 11.7 | yes | -10 | 3,500 |
| This work (2020) | 450 | 19 | 40 | 41.7 | 99.9905 | 4.4 | no | no | 1,300 |

Comments II (Reviewer 2):

*The description of measuring CHOCHO in the manuscript is limited, as the Fig. 2(b) only showed simultaneously detection of NO$_2$, IO, and O$_3$. It should be better if the authors could present a graph which contains 5 gas absorbers (NO$_2$, IO, O$_3$, CHOCHO, and H$_2$O) simultaneous retrieving. It should be noted that the concentrations of NO$_2$, O$_3$ shown in Fig. 2(b) were significantly higher than their concentrations in ambient air, even in polluted area. As the purpose of the manuscript is to present*

*an instrument for field application, it would be more persuasive for readers if a fitting example with low concentrations of gas absorbers could be provided.*

Answer: The spectrum presented in Figure 2 was obtained with synthetic air, explaining the high concentrations. High levels of $O_3$ were needed to produce IO from the $I_2$ source used, and high level of $NO_2$ were used to better visualize the different absorption components and identified correctly the structures of the spectra. Thus, the calibration and the intercomparison following the spectral fit description confirmed the well fitted spectra. Fortunately, the instruments came back from the field early June and we were able to measure the Glyoxal, $NO_2$ and $H_2O$ at lower concentrations levels. The Figure 2 of the manuscript was therefore modified to include CHOCHO and $H_2O$ spectra.

[Figure]

Comments III (Reviewer 2):

*The manuscript does not provide information about uncertainty of the instrumental measurements, as to the limit of detection (LOD), authors seems to confuse the concepts among LOD, sensitivity, and precision, because these three words appear alternately in Sect. 4.3.1. The using of these concepts needs to be clarified and revised in the manuscript.*

Answer: The manuscript describe a highly sensitive instrument in general (section 4.1) but reports the minimum detectable concentration as detection limit or limit of detection. Figure 7 shows the repeatability of the measurements over two tests while measuring for several hours the same zero-air sample in real conditions, therefore the term « precision » seems correctly used in this part of the manuscript. Nevertheless, Figure 7 was modified to better illustrate the repeatability of the instruments.

[Figure]

Additional comments (Reviewer 2):

1. *Line 56ff: References should not be quoted twice in the same sentence if it is already be written at the beginning. For example, "Venables et al. (2006) were ...... (Venables et al., 2006).":* corrected.
2. *Line 64: "Min et al 2016" → "Min et al. (2016)":* corrected.
3. *Line 65: "very high reflective mirrors[...]":* corrected.
4. *Line 130: Eq. (1) There are probably better ways to format the equations such that the size of the brackets is matched to the size of the arguments within the bracket:* corrected.
5. *Line 150: "Washenfelder et al. (2008) described[...]":* corrected.
6. *Line 152: "(e.g., helium versus air or nitrogen) [...]:* corrected.
7. *Such an approach to calculate mirror reflectivity has been proposed before (Venables et al., 2006) and has been used by previous studies (e.g., Duan et al., 2018). It would be better to reorganized the sentences in another way in the manuscript. In addition, did authors compare the difference between two reflectivity calibration methods based on their own IBBCEAS?*

Answer: We did the Rayleigh experiment using standard He gas (Messer, Helium 5.0, 99.999%) and standard $N_2$ gas (Air Liquide, AlphaGaz 2, 99.9999 %) cylinders, 5 µm Whatman® filters and taking into account the CCD dark noise. In addition, in between the field expeditions and the return of the instruments, we received a calibrator (Gas Standard Generator FlexStream™, Kin-Tek Analytical, Inc.) able to produce a stable $NO_2$ source. The sample is produced using a permeation tube of $NO_2$ (Kin-Tek ELSRT2W) calibrated at an emission rate of 115 ng min$^{-1}$ at 40 °C loaded into the calibrator. This type

of calibrator is ideally suited for creating trace concentration mixtures (from ppt to ppm). Despite those efforts, we were not satisfied by the results of this calibration method because of several arguments : one of them being the discrepancies between the Rayleigh cross sections provided by Min et al 2016 (empirical values) against the theoretical cross sections using equations provided by Thalman et al. (2014). The results of the experiment are shown in the Figure below : with Rayleigh curve we obtain 74.6 ppb of $NO_2$ using Min cross sections, and 98.0 ppb of $NO_2$ using Thalman cross sections, while the Kin-Tek $NO_2$ source was set at 49.6 ppb. The retrieved curve for matching $NO_2$ absorption cross sections is the blue one at the top, which also better match in shape the expected theoretical curve provided by the manufacturer. To confirm that the shape was correct, we compared the convoluted literature absorption cross sections of CHOCHO with the experimental data (which applies our experimental reflectivity curve) and we obtain a good matching, confirming that the shape of the curve is correct (see Fig SI – 3).

[Figure]

8. *Figure 2 (a): The text-label (i.e., Reflectivity) on the y-axis was covered.*

Answer: The Figure has been modified to add CHOCHO and $H_2O$ spectra as answered to Comment II.

9. *Line 202: In addition to the discrepancies at low $NO_2$ concentrations, obvious discrepancies measured by two instruments can also be observed at high $NO_2$ conditions, e.g., 18/10/01 - 09:00 and 19/07/19 – 05:45. Could authors provide an explanation about the phenomenon?*

Answer: We now used a Kintek $NO_2$ FlexStream[TM] in order to calibrate our IBBCEAS instrument. The non-linearity observed with the CLD technique was better explained in the manuscript: "In order to perform linearity tests, the previous $NO_2$ FlexStream[TM] calibrator was used to produced various concentrations of $NO_2$ covering a large range of concentrations, from few ppt to few ppb. Figure 4(b) shows the good linearity, from ppt to ppb range, of the IBBCEAS instrument with a slope of 1.015 ± 0.006 and a correlation factor of $R^2$ = 0.9996, confirming the validity of the calibration approach. The discrepancies observed between the IBBCEAS and the CLD techniques might be explain by positive and negative interferences on the CLD technique. While the system measures $NO_2$ directly, the CLD technique applies an indirect measurement of $NO_x$ from the oxidation of NO through a catalyzer, then in CLD, the $NO_2$ mixing ratio is obtained by subtracting the NO signal to the total $NO_x$ signal. Villena

et al. (2012), demonstrate that the interferences on a urban atmosphere for the CLD technique implied positive interferences when $NO_y$ species photolysis occurred, leading to an over-estimation of daytime $NO_2$ levels, while negative interferences were attributed to the VOCs photolysis followed by peroxyradical reactions with NO.''

10. *Figure 6 (top):The units of mixing ratio was missing :* corrected.
11. *Figure 7: The left Box-plot is not as useful as drawing a histogram which contains measured $NO_2$ concentrations when performing empty cavity measurements. Such a histogram can not only be used to show averages, but also be used to estimate LOD from the frequency number of histogram distribution.*

Answer: The Figure 7 has been modified as answered in Comment III to better show the precision or repeatability of the measurements.

12. *Line 247: A short discussion about the comparison shown in Table 1 is better than only presenting a Table without any explanation:* see answer to comment I.
13. *Line 308: "[...] sensor.The instruments[...]" -> "[...] sensor.The instruments[...]:* corrected.
14. *Line 324: As the inlet sampling line gets saturated in water vapor while passing through the ozone generator, did authors quantify the influence on CHOCHO measurements? For example, measure the CHOCHO standards with and without using ozone generator.*

Answer: The influence of water vapor while passing through the ozone generator on CHOCHO measurements was not tested. However, it was tested on the $NO_2$ measurements. Atmospheric measurements were done with and without fitting $H_2O$ to quantify the fitting interferences on $NO_2$. The Figure below shows the FIT results without, (left), and with, (right), $H_2O$ being included in the FIT routine. For this particular measurements, the results were giving 262.4 and 301.8 ppt of $NO_2$ and 4.7 and 4.4 ppm of $O_3$, respectively without and with the $H_2O$, leading, for this measurement, to an underestimation of 13 % on the $NO_2$ mixing ratio with the presence of 0.44 % humidity added by the $O_3$ production system in the sample line.

[Figure]

Additional remark (from the authors):

All the English mistake or the typo corrections suggested by the Reviewers have been corrected. Other changes coming from the opportunities of new experiments from the comments of the Reviewers were made. All the changes can be found in red in the manuscript and supplementary. Also, the following references where added or corrected:

- Duan, J., Qin, M., Ouyang, B., Fang, W., Li, X., Lu, K., Tang, K., Liang, S., Meng, F., Hu, Z., Xie, P., Liu, W., and Häsler, R.: Development of an incoherent broadband cavity-enhanced absorption spectrometer for in situ measurements of HONO and $NO_2$ Atmos Meas Tech, 11, 4531-4543, 2018.

- Liu, J., Li, X., Yang, Y., Wang, H., Wu, Y., Lu, X., Chen, M., Hu, J., Fan, X., Zeng, L., and Zhang, Y.: An IBBCEAS system for atmospheric measurements of glyoxal and methylglyoxal in the presence of high $NO_2$ concentrations, Atmos Meas Tech, 12, 4439-4453, 2019.

- Venables, D. S., Gherman, T., Orphal, J., Wenger, J. C., and Ruth, A. A.: High Sensitivity in Situ Monitoring of $NO_3$ in an Atmospheric Simulation Chamber Using Incoherent Broadband Cavity-Enhanced Absorption Spectroscopy, Environ Sci Technol, 40, 6758-6763, 2006.

- Villena, G., Bejan, I., Kurtenbach, R., Wiesen, P., and Kleffmann, J.: Interferences of commercial $NO_2$ instruments in the urban atmosphere and in a smog chamber, Atmospheric Measurement Techniques, 5, 149–159, https://doi.org/10.5194/amt-5-149-2012, https://www.atmos-meas-tech.net/5/149/2012/, 2012.

- Volkamer, R., Spietz, P., Burrows, J., and Platt, U.: High-resolution absorption cross-section of glyoxal in the UV–vis and IR spectral ranges, Journal of Photochemistry and Photobiology A: Chemistry, 172, 35–46, https://doi.org/10.1016/j.jphotochem.2004.11.011, https://linkinghub.elsevier.com/retrieve/pii/S1010603004005143, 2005.

- Wachsstock, D.: Tenua: the kinetics simulator for Java; http://bililite.com/tenua., http://bililite.com/tenua, 2007.